# Influenza virus recruits host protein kinase C to control assembly and activity of its replication machinery

Arindam Mondal[1†], Anthony R Dawson[1,2], Gregory K Potts[3], Elyse C Freiberger[4], Steven F Baker[1], Lindsey A Moser[5], Kristen A Bernard[5], Joshua J Coon[3,4], Andrew Mehle[1]*

[1]Medical Microbiology and Immunology, University of Wisconsin-Madison, Madison, United States; [2]Graduate Program in Cellular and Molecular Biology, University of Wisconsin-Madison, Madison, United States; [3]Department of Chemistry, University of Wisconsin-Madison, Madison, United States; [4]Department of Biomolecular Chemistry, University of Wisconsin-Madison, Madison, United States; [5]Department of Pathobiological Sciences, School of Veterinary Medicine, University of Wisconsin-Madison, Madison, United States

*For correspondence: amehle@wisc.edu

Present address: [†]School of Bioscience, India Institute of Technology Kharagpur, Kharagpur, India

Competing interests: The authors declare that no competing interests exist.

**Abstract** Influenza virus expresses transcripts early in infection and transitions towards genome replication at later time points. This process requires de novo assembly of the viral replication machinery, large ribonucleoprotein complexes (RNPs) composed of the viral polymerase, genomic RNA and oligomeric nucleoprotein (NP). Despite the central role of RNPs during infection, the factors dictating where and when they assemble are poorly understood. Here we demonstrate that human protein kinase C (PKC) family members regulate RNP assembly. Activated PKCδ interacts with the polymerase subunit PB2 and phospho-regulates NP oligomerization and RNP assembly during infection. Consistent with its role in regulating RNP assembly, knockout of PKCδ impairs virus infection by selectively disrupting genome replication. However, primary transcription from pre-formed RNPs deposited by infecting particles is unaffected. Thus, influenza virus exploits host PKCs to regulate RNP assembly, a step required for the transition from primary transcription to genome replication during the infectious cycle.
DOI: https://doi.org/10.7554/eLife.26910.001

## Introduction

Influenza virus infections initiate with a burst of gene expression from pre-formed RNPs deposited by the incoming viral particles. Primary transcription is followed by replication of the genome and subsequent transcription of the replicated genome, further increasing gene expression. This transition from transcription to replication requires the de novo assembly of RNPs and is absolutely required for successful infection and the production of infectious progeny. Viral product have been proposed to regulate this transition: NEP has been shown to shift the viral polymerase towards replication; svRNAs are thought to associate with the viral polymerase and promote copying of full-length genomic RNA; and newly synthesized polymerase proteins have been proposed to stimulate replication in *trans* (*Jorba et al., 2009*; *Perez et al., 2012*, *2010*; *Robb et al., 2009*; *York et al., 2013*). Nonetheless, the mechanisms regulating upstream events of RNP assembly and the host factors contributing to this coordinated shift from transcription to RNP assembly and genome replication are largely unknown.

The influenza virus RNP is a double helical structure containing the viral polymerase and repeating NP subunits coating each of the eight genomic RNAs (*Arranz et al., 2012*; *Klumpp et al., 1997*;

**eLife digest** To be able to multiply, the influenza virus needs to enter the cells of its host and trick them into copying the virus' genetic information and assembling new virus particles. The genetic information of the virus is stored in molecules of ribonucleic acid (RNA) and encodes several viral proteins that are involved in making the new virus particles. These proteins include an enzyme known as the viral polymerase and a "nucleoprotein". The viral polymerase copies the RNA and then the nucleoprotein binds to the new RNA to protect it until it is packaged into new virus particles. Many nucleoprotein units assemble into long chains that coat the whole length of the RNA, but it is not yet known exactly how this process is controlled.

In cells, other enzymes known as kinases are able to alter the activities of many proteins by modifying the structures of proteins by a process called phosphorylation. Influenza nucleoprotein was previously shown to be phosphorylated. It is therefore possible that the influenza virus may use phosphorylation to control the assembly of nucleoproteins into chains along the RNA. However, the virus' RNA does not encode any kinase enzymes of its own, so it must rely on kinases from its host cell.

Human cells produce many kinase enzymes that can be grouped into several different protein families. Mondal et al. studied the role of the protein kinase C family in making new virus particles. The experiments show that modifying the members of this protein family to be permanently active causes the viral nucleoprotein to be phosphorylated at two specific sites on the protein. This regulates the assembly of the nucleoproteins into long chains on the RNA, and ultimately promotes the production of new virus particles. Closer examination revealed that this effect was primarily down to one specific kinase known as PKCδ. The virus was less able to multiply in human lung cells that were missing PKCδ – specifcially because the formation of nucleoprotein chains was no longer regulated – and these cells produced lower quantities of virus proteins.

Taken together, these findings show that kinases produced by host cells can control the ability of viruses to replicate by modifying the viral nucleoproteins. In the future, it may be possible to develop new drugs that target PKCδ and other cellular factors the virus needs to help treat influenza infections.

DOI: https://doi.org/10.7554/eLife.26910.002

*Moeller et al., 2012*; *Pons et al., 1969*). The viral polymerase, a heterotrimer composed of the subunits PB1, PB2 and PA, is located at one end of the RNP where it binds both the 5' and 3' genomic termini. This RNP performs both transcription and replication. Transcription of viral mRNAs occurs via a 'cap-snatching' mechanism, beginning immediately following nuclear import of the incoming RNPs and continuing throughout infection (*Bouloy et al., 1978*; *Plotch et al., 1981*). Replication occurs at later time points when RNPs direct synthesis of a positive-sense complementary RNA (cRNA) intermediate that templates replication of the negative-sense viral RNA genome (vRNA) (*Hay et al., 1977*). Importantly, this replication requires the assembly of RNPs containing newly synthesized polymerase, NP, and either cRNA (cRNPs) or vRNA (vRNPs) (*Barrett et al., 1979*; *Vreede et al., 2004*). To fully coat the genome, NP forms homo-oligomers and binds RNA in a sequence-independent fashion. These same properties cause NP to oligomerize spontaneously and bind non-specifically to cellular RNAs (*Baudin et al., 1994*; *Prokudina-Kantorovich and Semenova, 1996*; *Zhao et al., 1998*). Therefore, control of NP oligomerization and RNP assembly are key regulatory steps as the infectious cycle progress towards genome replication.

Influenza virus NP oligomerizes by inserting a small 'tail loop' (aa 402–428) into the binding groove of a neighboring protomer (*Ng et al., 2008*; *Ye et al., 2006*). NP binds RNA via a large basic surface and is thought to encapsidate the nascent RNA genome concomitant with its synthesis, hence a continuous supply of RNA-free monomeric NP is required for assembly into RNA-bound RNPs and replication of the viral genome (*Beaton and Krug, 1986*; *Ng et al., 2008*; *Shapiro and Krug, 1988*; *Vreede et al., 2004*). We and others reported that phosphorylation at the homotypic interface inhibits NP oligomerization during both influenza A and B virus replication (*Chenavas et al., 2013*; *Hutchinson et al., 2012*; *Mondal et al., 2015*; *Turrell et al., 2015*). Specifically, phosphorylation or phospho-mimetics at residue S165 in the groove or S407 in the tail loop

drives influenza A NP towards a monomeric state, prevents RNP assembly, and severely impairs viral replication (*Mondal et al., 2015*). NP mutants lacking key phospho-sites are also defective in supporting influenza polymerase activity and virus replication, and in some cases result in NP hyper-oligomerization (*Mondal et al., 2015*; *Turrell et al., 2015*). Thus, both hyper- and hypo-phosphorylation of NP is deleterious suggesting that the reversible phosphorylation of NP must be carefully balanced to enable recruitment of oligomerization-competent NP to sites of genome replication and ultimately incorporation into growing RNPs. Influenza virus does not encode a kinase, therefore the phospho-regulation of NP must be performed by host enzymes.

Here we identify the protein kinase C (PKC) family, and PKCδ in particular, as host kinases that control RNP assembly by phospho-regulating NP oligomerization and subsequently impact the transition from gene expression to genome replication. We show that PKC activity disrupts influenza virus polymerase function and that polymerase-associated PKCδ specifically phosphorylates NP. PKCδ is recruited by the polymerase subunit PB2 and targets key residues at the tail loop:groove interface to regulate NP oligomerization. Knockout of PKCδ in human lung cells decreased NP phosphorylation during infection and significantly reduced viral gene expression and production of infectious progeny. As *de novo* formation of RNPs is required for genome replication and the amplification of viral gene expression, these findings predict that PKCδ is important at late stages of infection. Indeed, primary transcription at early time points was unaffected in PKCδ knockout cells whereas the transition to genome replication at later time points was severely impaired. Thus, influenza virus exploits host PKCδ to regulate the ordered assembly of RNPs enabling the resultant transition from gene transcription to genome replication.

## Results

### Constitutively active PKC impairs viral polymerase activity by phosphorylating NP

Both activators and inhibitors of the PKC family have been shown to modulate influenza virus replication (*Hoffmann et al., 2008*; *Kistner et al., 1989*). More recently we demonstrated that activating PKCs with phorbol-12-myristate-13-acetate (PMA) stimulates NP phosphorylation and inhibits its oligomerization (*Mondal et al., 2015*). As NP oligomerization and RNP assembly are required for replication of the viral genome, we undertook a targeted approach to investigate the role of PKCs in regulating influenza polymerase activity.

The PKC family consists of at least eleven different members which can be divided into classical (α, β1, β2, γ), novel (δ, ε, η, θ) and atypical (ι/λ, ζ) isoforms based on their structure and co-factor requirements. To test the ability of PKC isoforms to phosphorylate NP and impact polymerase function, we performed polymerase activity assays in cells co-expressing a panel of PKC variants. Influenza polymerase activity was reconstituted in cells by expressing the trimeric polymerase, NP, and a vRNA-like reporter encoding luciferase. The viral reporter is replicated and transcribed only in the presence of a functional polymerase and NP, and serves as a proxy for RNP formation (*Figure 1—figure supplement 1*). PKCs were expressed as constitutively active truncations containing the C-terminal catalytic domain (PKC-CAT), but lacking the regulatory domains (*Soh and Weinstein, 2003*). In a cell, PKCs are synthesized in an inactive conformation and are activated upon binding phosphatidylserine, and in most cases also require binding to diacylglycerol, $Ca^{2+}$ or phosphatidylinositol 4,5-bisphosphate (*Antal and Newton, 2014*). Using the constitutively active forms eliminated variability that may arise from the distinct second messenger activators required by different PKC family members. Expressing active PKCβ2, PKCδ, PKCθ and PKCη considerably reduced polymerase activity with respect to the empty vector control. While PKCβ2 and PKCη showed 60–80% decrease, PKCδ and PKCθ abrogated polymerase activity completely to background levels (*Figure 1A*). PKCε and PKCα showed moderate, but statistically significant reductions, whereas the remaining isoforms caused minimal changes in polymerase activity, or even minor increases in activity. Western blotting showed that slower migrating forms of NP appeared in conditions where polymerase activity was inhibited. Phosphatase treatment confirmed that these slower migrating species resulted from NP hyper-phosphorylation and quantification revealed a strong correlation between increased NP phosphorylation and decreased polymerase activity (*Figure 1—figure supplement 2A–B*). Blotting also showed comparable expression of all PKC isoforms and detected previously described minor bands

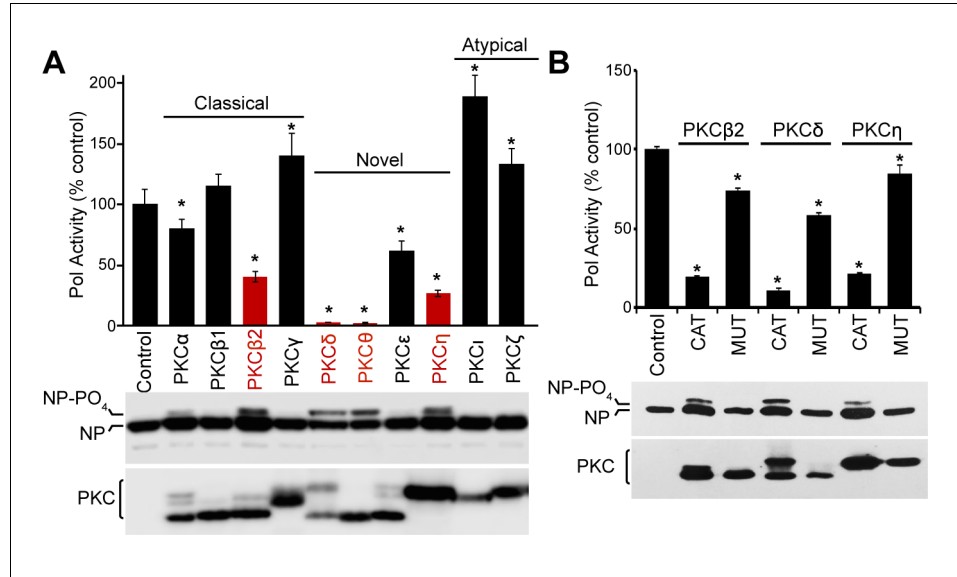

**Figure 1.** Constitutively active PKC phosphorylates NP leading to impaired influenza virus polymerase activity. (**A**) Expression of constitutively active PKC impairs influenza virus polymerase activity. Polymerase activity assays were performed in 293T cells in the presence or absence of the catalytic domains from classical, novel or atypical PKC isoforms. Data were averaged and normalized to the empty vector control. NP and PKC were detected by western blotting whole cell lysate. A hyper-phosphorylated form of NP was detected in some conditions. (n=3 ± standard deviation, *p<0.05 one-way ANOVA when compared to the empty vector control). (**B**) Polymerase activity assays were performed in the presence of PKC catalytic domains, catalytically inactive mutants, or empty vector controls. Polymerase activity and protein expression were analyzed as in (**A**).
DOI: https://doi.org/10.7554/eLife.26910.003
The following figure supplements are available for figure 1:

**Figure supplement 1.** Polymerase activity assays.
DOI: https://doi.org/10.7554/eLife.26910.004
**Figure supplement 2.** PKCs hyper-phosphorylate NP and only catalytically active enzymes inhibit viral polymerase activity.
DOI: https://doi.org/10.7554/eLife.26910.005

due to differential post-translational modifications (*Soh and Weinstein, 2003*). Subsequent experiments were focused on PKCβ2, PKCδ, and PKCη, but not PKCθ as its expression is heavily restricted to skeletal muscle and cells of the immune system and not the lung epithelial cells where influenza virus primarily replicate (*Zhang et al., 2013*).

To determine if kinase activity from different PKC isoforms drives polymerase activity inhibition, polymerase activity assays were repeated in the presence of inactive PKC mutants with single amino acid changes in their catalytic domain. Whereas the catalytic domains of PKCβ2, PKCδ and PKCη inhibited polymerase activity, this phenotype was significantly reduced for the inactive mutants (*Figure 1B* and *Figure 1—figure supplement 1*). Moreover, NP was not hyper-phosphorylated in the presence of the inactive mutants, suggesting that specific PKC isoforms inhibit influenza polymerase activity by causing the phosphorylation of NP.

## PB2 stabilizes interactions between PKCδ and NP

PKCs can function directly by phosphorylating a target or indirectly by activating downstream kinases that then phosphorylate the target protein. To determine if the effects of PKCs on polymerase activity were direct or indirect, we performed binding assays to test if inhibitory isoforms of PKC associate with proteins in the viral RNP. Whereas NP is phosphorylated when PKCs are expressed and this correlates with the inhibitory phenotype, co-immunoprecipitations failed to detect stable interactions between NP and PKC when co-expressed in 293T cells (*Figure 2A*). This was consistent with the transient interactions frequently observed between kinase and substrate. Surprisingly, NP

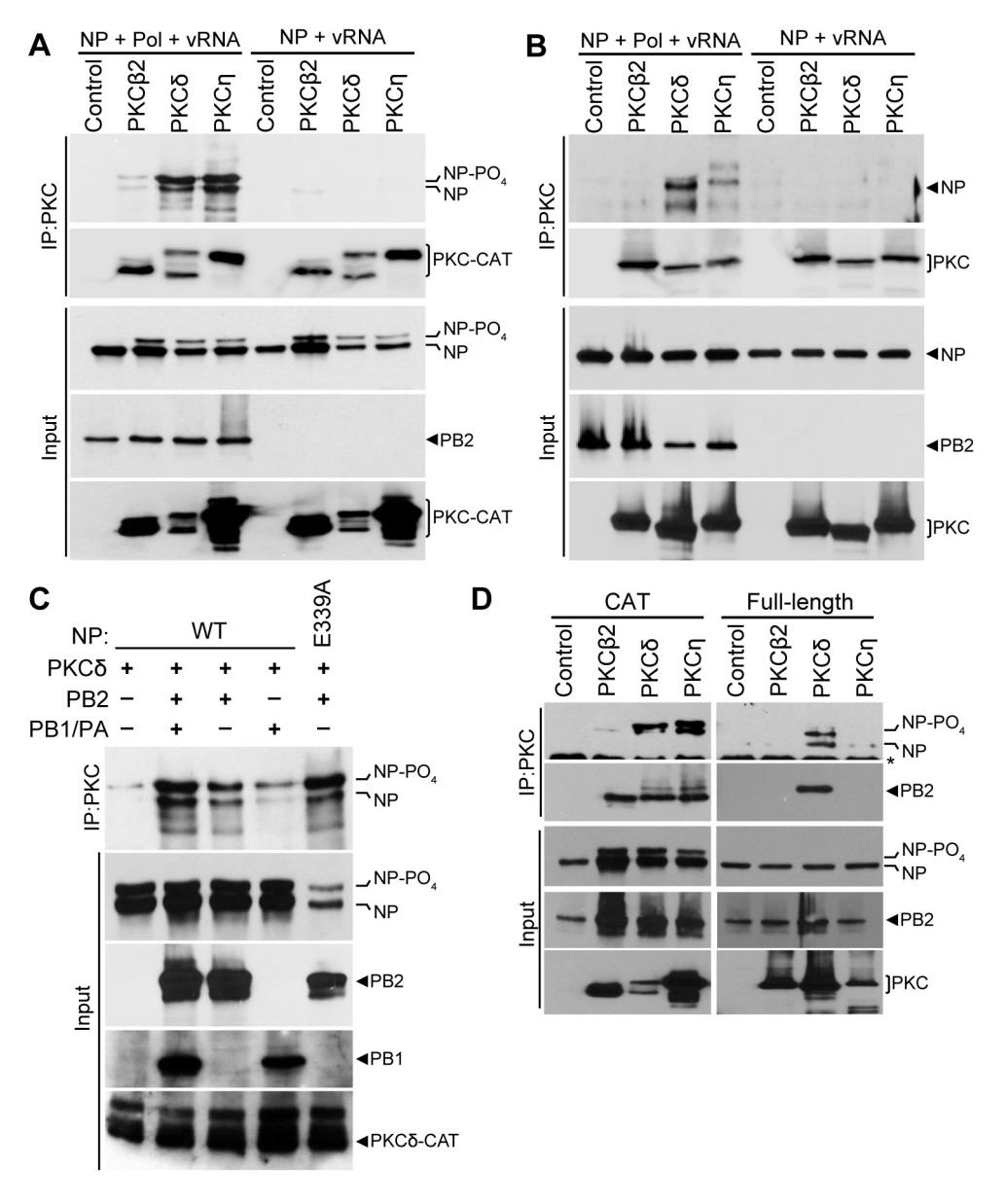

**Figure 2.** The polymerase subunit PB2 bridges stable interactions between PKCδ and NP. (**A**) Catalytic fragments or (**B**) full-length PKC isoforms were co-expressed with NP and vRNA in the presence or absence of the viral polymerase (Pol). PKC was immunoprecipitated from cell lysates and precipitated PKC and co-precipitated NP were detected by western blot. Input proteins were detected by blotting whole cell lysates for NP, PB2 and PKC. (**C**) To determine which polymerase subunits facilitate PKC binding, interaction assays were performed as above on cells expressing PKCδ, WT or oligomerization-defective NP (E339A), and the indicated combinations of polymerase proteins. (**D**) The minimal components needed for complex formation were tested by immunoprecipitating catalytic fragments or full-length PKCs from lysates where NP and PB2 were co-expressed. Immunoprecipitated and input proteins were detected by western blot.

DOI: https://doi.org/10.7554/eLife.26910.006

was efficiently co-precipitated with catalytic fragments of PKCδ and PKCη when they were co-expressed with the other components of the RNP (i.e. viral polymerase and vRNA) (*Figure 2A*). PKCβ2 co-precipitated only limited amounts of NP. Notably, NP co-precipitated by PKCδ and PKCη was highly enriched for the hyper-phosphorylated form relative to its abundance in total cell lysate.

To ensure specificity of these interactions, experiments were repeated using full-length PKC isoforms (*Figure 2B*). Again, NP was co-precipitated by full-length PKC when the polymerase and vRNA were co-expressed. Differences in NP co-precipitation were not due to differences in immunoprecipitation of the different PKC isoforms, as each PKC immunoprecipitated with equivalent efficiency relative to its expression in the cell lysate (*Figure 2A–B*). Additionally, NP showed a clear preference for interaction with PKCδ, although PKCη also co-precipitated minor amounts of NP.

Interactions between NP and PKC isoforms were enhanced in the presence of the viral polymerase and vRNA (*Figure 2A–B*). This enhanced interaction was still observed when the vRNA template was excluded from transfections or when RNaseA was included during the immunoprecipitation (not shown), suggesting that the viral polymerase is sufficient to mediate the interaction between NP and PKC. We determined which proteins of the heterotrimeric polymerase are essential for NP-PKCδ interactions (*Figure 2C*). Reconstituting the complete polymerase by expressing PB1, PB2 and PA enabled strong co-precipitation of NP by PKCδ. Only minor amounts of NP were co-precipitated in the absence of the polymerase. Interestingly, co-expression of PB2 alone was sufficient to ensure significant co-precipitation of NP by PKCδ, whereas co-expression of PB1 and PA resulted in limited co-precipitation similar to that observed in the absence of the polymerase. Furthermore, the oligomerization defective NP E339A was co-precipitated by PKCδ in the presence of PB2, suggesting NP monomers also participate in this interaction. To obtain further evidence for this interaction and identify the most relevant PKC isoforms, we performed a co-immunoprecipitation experiment in cells expressing NP, PB2 and either catalytic or full-length PKC isoforms (*Figure 2D*). PB2 bridged interactions between NP and the catalytic domains of PKCδ and PKCη. In the context of full length protein, PKCδ showed the strongest interactions with NP and PB2. Our data suggested that PB2 anchors a hetero-oligomeric NP:PB2:PKCδ complex. NP present in this immuno-precipitated complex was significantly enriched for the hyper-phosphorylated form, suggesting that PB2 facilitates a functional interaction between NP and activated PKC resulting in NP phosphorylation.

## PKC phosphorylates NP at the tail loop:groove interface and blocks oligomerization

Phosphorylation of NP at the homotypic interface inhibits oligomerization (*Chenavas et al., 2013*; *Mondal et al., 2015*; *Turrell et al., 2015*). Our data implicate PKCs as the host kinases that phospho-regulate NP oligomerization. We therefore assessed the ability of PKC isoforms to directly phosphorylate NP. PKC catalytic domains were immunopurified from cell lysates and used in *in vitro* kinase assays with recombinant NP (*Figure 3A*). All three PKC isoforms that inhibited polymerase activity, i.e. PKCβ2, PKCδ and PKCη, phosphorylated NP. By contrast, PKCε, which exhibited modest inhibition of polymerase activity, showed no specific kinase activity and its activity was similar to that in the negative control. To demonstrate specificity, the PKC inhibitor 1-(5-Isoquinolinesulfonyl)−2-methylpiperazine (H7) was included in kinase reactions. H7 almost completely eliminated NP phosphorylation (*Figure 3B*), compared to the vehicle-only control that did not impact NP phosphorylation. These kinase assays showed that select PKC isoforms, and not other fortuitously co-precipitating kinases, directly phosphorylated NP *in vitro*.

If PKC regulates NP:NP interactions, phospho-sites at this interface are likely PKC targets. But, our *in vitro* kinase assays were performed with purified recombinant NP, which is a complex mixture of monomeric and multimeric species in equilibrium where many of the key phospho-regulatory sites may be concealed by NP:NP interactions (*Figure 3—figure supplement 1A*) (*Mondal et al., 2015*). To gain additional insight into the impact of PKC on NP oligomerization, we performed *in vitro* kinase assays with either wild-type (WT) NP or the oligomerization-defective mutant NP E339A. NP E339A purified exclusively as a monomer with the phospho-sites S165 and S407 solvent exposed compared to NP oligomers where they are concealed at the protein:protein interface (*Figure 3—figure supplement 1A*) (*Chenavas et al., 2013*; *Ye et al., 2013*; *Ye et al., 2006*). Kinase assays again demonstrated that NP is phosphorylated by the catalytic domains of PKCβ2, PKCδ and PKCη. Compared to the WT NP control, phosphorylation of NP E339A was markedly increased by all three PKC isoforms, supporting a model where PKC targets sites at the NP:NP interface (*Figure 3C*).

Multiple phospho-sites have recently been mapped to the NP:NP interface, including S165 in the groove and S402, S403, S407, S413 in the tail loop (*Hutchinson et al., 2012*; *Mondal et al., 2015*). Our prior mutagenesis showed that NP S165 and S407 are the crucial phospho-sites that regulate homo-oligomerization and RNP formation (*Mondal et al., 2015*). We asked whether PKCs selectively

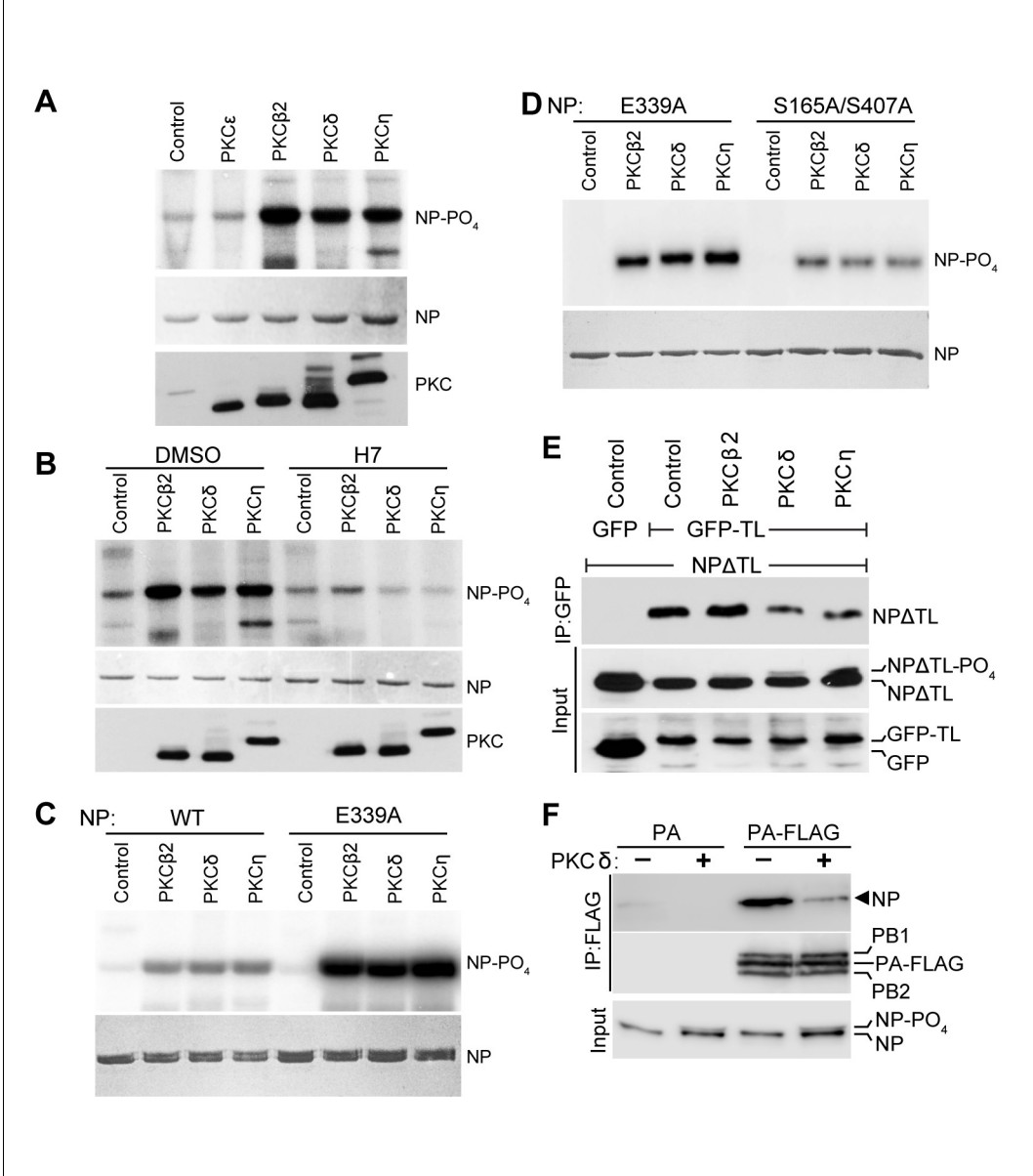

**Figure 3.** PKC regulates NP oligomerization by phosphorylating residues at the homotypic interface. (**A**) Purified PKC phosphorylates NP *in vitro*. Catalytic fragments of PKC were immunoprecipitated from 293T cells and used for *in vitro* kinase assays with recombinant NP. Immunoprecipitates from cells transfected with empty vector were used as a negative control. Kinase reactions were resolved by SDS-PAGE followed by Coomassie staining to monitor NP levels and autoradiography to detect phosphorylation. PKC levels were measured by western blotting a portion of the immunoprecipitate. (**B**) *In vitro* kinase assays were performed with purified PKC and NP supplemented with the PKC inhibitor H7 or the vehicle control. Reaction products were analyzed as described in (**A**). (**C–D**) *In vitro* kinase assays were performed to identify NP residues phosphorylated by PKC, using (**C**) WT and oligomerization-defective NP (E339A) as substrate or (**D**) oligomerization-defective NP (E339A) and a double-alanine mutant of NP (S165A/S407A) as substrate. (**E**) PKC activity reduces NP tail loop:groove interactions. An NP deletion mutant lacking the tail loop (NPΔTL) and a tail loop fused to eGFP (eGFP-TL) were co-expressed in 293T cells in the presence or absence of exogenous PKCδ. NP tail loop:groove interactions were monitored by immunoprecipitating eGFP-TL and measuring co-precipitating NPΔTL by western blot. Expression levels of interacting partners were analyzed by blotting total proteins. (**F**) PKCδ activity impairs RNP assembly. RNPs were reconstituted in cells expressing a vRNA, PB1, PB2, NP and PA or PA-FLAG in the presence or absence of exogenous PKCδ. RNPs were captured by FLAG immunoprecipitation. Co-precipitating and input proteins were detected by blotting NP or the polymerase proteins.

DOI: https://doi.org/10.7554/eLife.26910.007

*Figure 3 continued on next page*

*Figure 3 continued*

The following figure supplements are available for figure 3:

**Figure supplement 1.** PKC phosphorylates NP S165 and S407.
DOI: https://doi.org/10.7554/eLife.26910.008

**Figure supplement 2.** Protein interaction assays to measure NP:NP association and RNP assembly.
DOI: https://doi.org/10.7554/eLife.26910.009

phosphorylate NP S165 and S407 and whether this regulates oligomerization. A double mutant eliminating both key phospho-sites, NP S165A/S407A, was purified and used as substrate in *in vitro* kinase assays. NP S165A/S407A purifies as a monomer due to the loss of important inter-subunit hydrogen bonds involving these serines (*Figure 3—figure supplement 1A*) (*Mondal et al., 2015*). Phosphorylation was significantly reduced for NP S165A/S407A compared to the high levels detected for the monomer NP E339A (*Figure 3D*). This defect was most pronounced for PKCδ and PKCη, where mutation of NP S165 and S407 reduced phosphorylation by almost 50% (*Figure 3—figure supplement 1B*), confirming that these sites are major targets for PKC-mediated phosphorylation. Residual phosphorylation detected on the NP S165A/S407A mutant may reflect phosphorylation at one of the other previously identified phospho-sites (*Hutchinson et al., 2012*; *Mondal et al., 2015*), and our quantitative mass spectrometry suggest that some of these sites could also be targets of PKCs (see below).

We subsequently tested the functional consequences of PKC-mediated phosphorylation in cells. Each NP protomer contains a tail loop and binding groove, and can thus both insert into a growing NP chain and then receive the next incoming subunit. To understand the specific steps of NP self-association impacted by PKC phosphorylation, the complex multivalent nature of NP oligomerization was simplified to a binary interaction in our tail loop:groove interaction assay (*Mondal et al., 2015*). Binding was measured between an NP deletion lacking the tail loop (NPΔTL), which cannot oligomerize on its own, and the NP tail loop fused to green fluorescent protein (GFP-TL) (*Figure 3—figure supplement 2*). Binding partners were expressed in cells either with constitutively active PKC isoforms or controls (*Figure 3E*). In control experiments, GFP-TL co-precipitated NPΔTL, recapitulating NP:NP interactions. Expression of PKCδ or PKCη severely reduced NP:NP interactions and co-precipitation of NPΔTL. Moreover, this loss of function in the presence of PKCδ and PKCη was again associated with the appearance of minor hyper-phosphorylated forms of NP. Whereas PKCβ2 can phosphorylate NP in an isolated kinase assay, expression of PKCβ2 in cells did not change the amount of co-precipitated NPΔTL. This disparity perhaps reflects the enhanced specificity of substrate:kinase interactions in cells or the involvement of other cellular co-factors that affect the functional impact of NP phosphorylation. Finally, we assessed the impact of PKCδ on RNP assembly by measuring the amount of NP co-precipitated with PA during a polymerase activity assay (*Figure 3—figure supplement 2*). As NP does not interact directly with PA, co-precipitation can only occurs in the context of an RNP. NP was specifically co-precipitated by PA-FLAG, yet these interactions were nearly eliminated when PKCδ was co-expressed (*Figure 3F*). Notably, co-precipitation of the other polymerase subunits PB1 and PB2 was unaffected by PKCδ expression, suggesting that phosphorylation by PKCδ selectively impairs RNP assembly but not polymerase trimer formation. Together these results provide strong evidence that PKCδ and PKCη disrupt influenza polymerase activity by phospho-regulating NP tail loop:grove interactions, and thus have the potential to control NP oligomerization and RNP formation in cells.

## Activated PKCδ associates with the viral polymerase during infection

To confirm the biological importance of the NP:PB2:PKC complex, we examined its formation and activity during infection in the presence of endogenous levels of PKC. We focused on PKCδ given that NP exclusively interacts with full-length PKCδ (*Figure 2B,D*), PKCδ phospho-regulates NP oligomerization and RNP assembly (*Figure 3*), and that PKCδ is abundantly expressed in human lung tissue, typical sites of influenza virus replication, and the human lung epithelial A549 cells used here (*Goldberg and Steinberg, 1996*). A549 cells were infected with influenza virus, influenza virus encoding FLAG-tagged PB2, or mock treated. PB2-containing complexes were purified by anti-FLAG immunoprecipitation and blotted with anti-PKCδ antibody. Endogenous PKCδ specifically co-

precipitated with PB2-FLAG (*Figure 4A*). The reciprocal immunoprecipitation performed on infected cell lysates showed a similar interaction; PB2 specifically co-precipitated with endogenous PKCδ, compared to background amounts precipitated by a non-specific control (*Figure 4B*). These data demonstrated interactions between PKCδ and PB2 during infections and validate results from our transfection assays showing that PB2 anchors interactions between NP and active PKC. This conclusion raised the possibility that PB2 isolated from infected cells contained PKC-specific kinase activity. To test this, PB2 complexes purified from infected cell lysate were used as a source of both kinase and substrate for *in vitro* phosphorylation. *In vitro* kinase assays performed with immuno-precipitated PB2 complexes revealed specific phosphorylation of NP, which was eliminated when the PKC inhibitor H7 was included in the reaction (*Figure 4C*). Moreover, treatment of the immunoprecipitated complex with RNase A markedly increased phosphorylation, suggesting that the lower-order NP released by RNase A treatment rather than the oligomerized RNA-associated form was a better substrate for the kinase activity associated with the complex. This parallels our prior results showing enhanced phosphorylation of monomeric NP (*Figure 3C*). Note that these kinase reactions did not include any exogenous second messenger activators of PKC, suggesting that PB2 is associated with activated PKCδ. Phospho-proteomics provided further evidence that PKCδ is activated during influenza. We identified peptides from PKCδ containing phosphorylated serines at residues 302 and 304 (*Table 1*), an autocatalytic marker of PKCδ activation (*Durgan et al., 2007*). Combined, these data

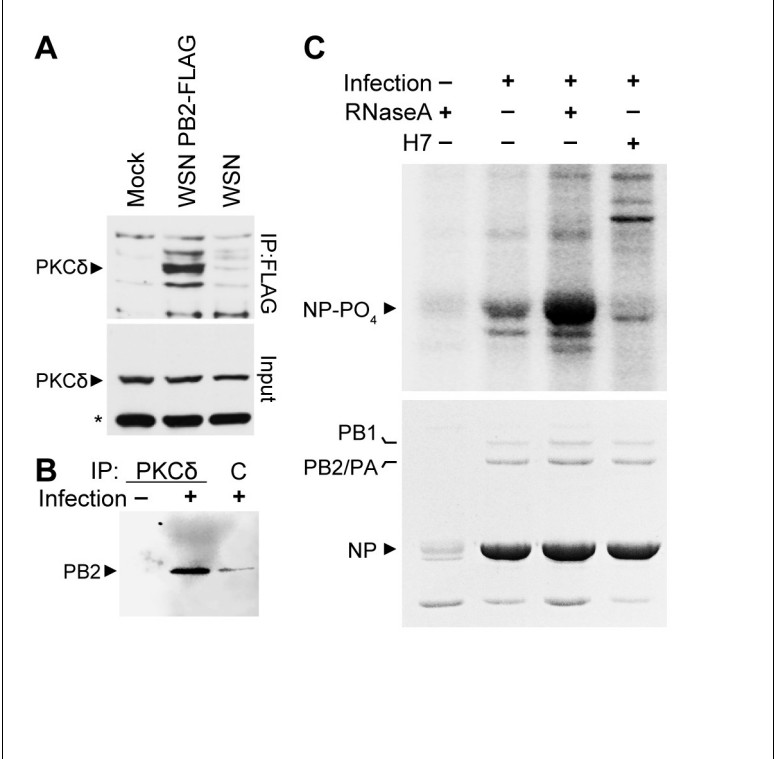

**Figure 4.** Polymerase-associated PKCδ directs NP phosphorylation during infection. (**A**) A549 cells were infected with WSN, WSN PB2-FLAG, or mock treated. Cell lysates were subjected to anti-FLAG immunoprecipitation and blotted for PKCδ. Whole cell lysate were also probed for PKCδ. *=nonspecific cellular protein. (**B**) Infected or mock-treated A549 cell lysates were immunoprecipitated with anti-PKCδ antibodies or a non-specific control and analyzed by blotting for PB2. (**C**) PB2-FLAG was immunoprecipitated from infected A549 cells. Immunoprecipitates were used as a source of both kinase activity and substrate for *in vitro* kinase assays. Where indicated, immunoprecipitated complexes were treated with RNaseA or H7. Reactions were analyzed by SDS-PAGE followed by Coomassie staining and autoradiography. IgG was detected by Coomassie staining immunoprecipitated samples, including heavy chain in the uninfected control which co-migrates with NP.
DOI: https://doi.org/10.7554/eLife.26910.010

**Table 1.** PKCδ is activated during influenza virus infection.

| Position | Phospho (STY) Probabilities | Position in peptide | Charge | Mass error [ppm] |
|---|---|---|---|---|
| S302 | RS(0.52)DS(0.378)AS(0.097)S(0.005)EPVGIYQGFEK | 2 | 3 | 0.87569 |
| S304 | RS(0.003)DS(0.995)AS(0.002)SEPVGIYQGFEK | 4 | 2 | 0.51199 |

Phosphoproteomic analysis was performed by mass spectrometry on lysates prepared from A549 cells at 6 hr post-inoculation. Phosphopeptides corresponding to PKCδ autocatalytic modifications at residues S302 and S304 were identified. Localization probabilities for each phosphosite are shown in parentheses.

DOI: https://doi.org/10.7554/eLife.26910.011

provide multiple lines of evidence that the viral polymerase protein PB2 anchors active PKCδ during infection.

## NP phosphorylation and influenza virus replication are impaired in PKCδ-deficient cells

We generated PKCδ-deficient human lung A549 cells to study NP phospho-regulation during infection. Two independent clonal cell lines with nonsense mutations in both *PRKCD* alleles were created using the CRIPSR-Cas9 system (*Figure 5—figure supplement 1A–B*). PKCδ protein expression was completely abolished in the knockout cells, while very low levels of PKCβ2 were detected by the cross-reactive antibody (*Figure 5A*, *Figure 5—figure supplement 1C*). Despite the loss of PKCδ, both lines showed regular morphology and grew similar to the parental A549 cells. Regardless, care was taken to use early passage cells to avoid compensatory changes in gene expression or phenotypic drift that may arise over time.

To test if PKCδ is responsible for NP phosphorylation during infection, we employed quantitative mass spectrometry to measure the extent of NP phosphorylation in the knockout cell lines (*Merrill and Coon, 2013*; *Richards et al., 2015*). NP was purified from infected WT or *PRKCD*$^{-/-}$ cell lines and analyzed by targeted mass spectrometry (*Mondal et al., 2015*; *Peterson et al., 2012*). Phosphorylation was detected at sites in the tail loop (S402, S403, S407, S413), binding groove (S165), and the body domain (T378) of NP, in agreement with prior results (*Figure 5B–C*) (*Hutchinson et al., 2012*; *Mondal et al., 2015*). Compared to the WT cells, NP phosphorylation was considerably reduced in the *PRKCD*$^{-/-}$ cells (*Figure 5C*). This was especially true at the key regulatory positions of NP S165 and S407, where phosphorylation levels decreased by 60–80%. In two out of three replicates, pS407 was barely detected in samples from knockout cells. By contrast, the relative abundance of phosphorylation at T378 in the body was only marginally affected. These data illustrate that PKCδ is an important kinase targeting phospho-regulatory sites at the site of NP:NP interface. However, as phosphorylation was not completely ablated in the absence of PKCδ, these results also suggest at least some functional redundancy for NP phosphorylation, possibly fulfilled by other PKC isoforms.

Given the reduction of NP phosphorylation at S165 and S407 in PKCδ knockout cell lines and the importance of these phosphorylation sites in virus replication, we predicted that viral gene expression and replication should be impaired in these cells. To test this, WT or *PRKCD*$^{-/-}$ A549 cells were infected with influenza reporter viruses (PASTN) based on the lab-adapted strain WSN or a primary isolate from the 2009 pandemic (CA04) (*Karlsson et al., 2015*; *Tran et al., 2013*; *Tran et al., 2015*). Gene expression was reduced for both viruses by ~60% in the *PRKCD*$^{-/-}$−1 cell line and ~30% in the *PRKCD*$^{-/-}$−2 line (*Figure 5D*). This PKCδ-dependent defect was not simply a generic reduction in viral entry or gene expression, as gene expression from a West Nile virus replicon did not differ significantly between WT and knockout cells (*Figure 5D*).

PKCs have been previously implicated in the process of influenza virus entry, especially PKCβ2 (*Root et al., 2000*; *Sieczkarski et al., 2003*). To separate this role in entry from our functional analysis, we bypassed normal HA-mediated attachment and entry by performing infections with an influenza reporter virus pseudotyped with the vesicular stomatitis virus glycoprotein (FVG-R) (*Watanabe et al., 2003*). Flow cytometry confirmed that entry of the pseudotyped influenza virus or *bona fide* vesicular stomatitis virus was similar in the presence or absence of PKCδ (*Figure 5—figure supplement 2A*). Even with the pseudotyped FVG-R virus, influenza gene expression was impaired in the knockout cell lines, indicating that PKCδ is important post-entry for gene expression and

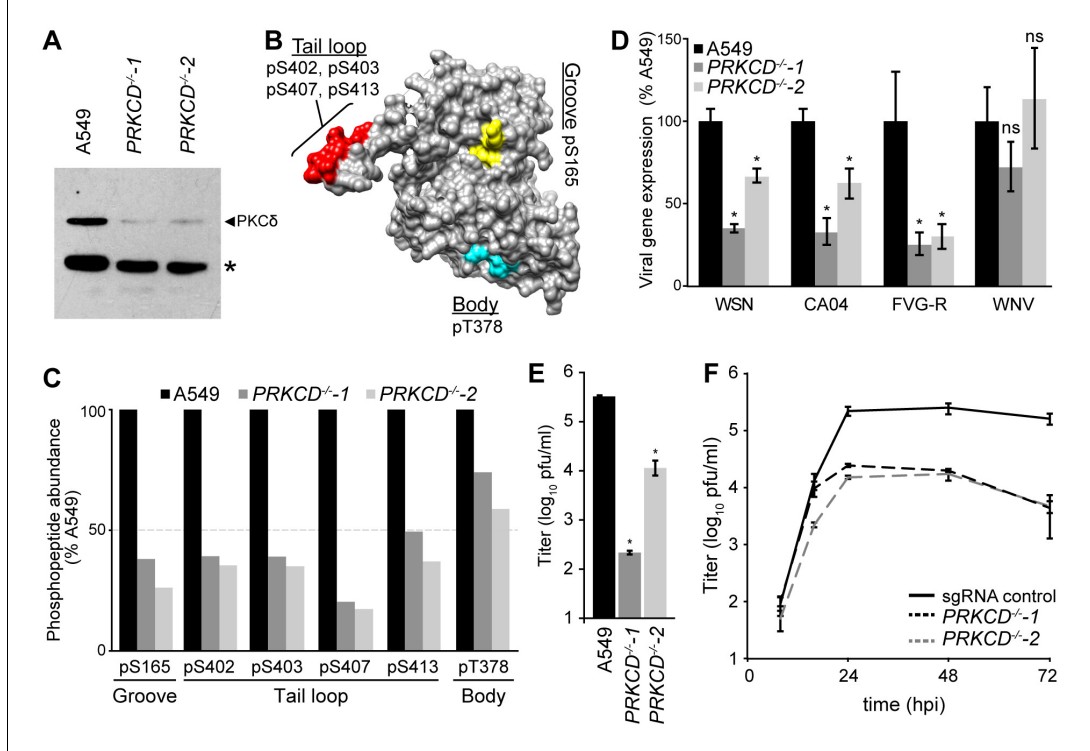

**Figure 5.** Knockout of PKCδ impairs NP phosphorylation and virus replication. (**A**) Ablation of PKCδ expression in *PRKCD*-knockout A549 cells was confirmed by western blotting cell lysates from WT and two clonal knockout cell lines. Residual signal is due to low-level antibody cross-reactivity to PKCβ2 present in these cells (***Figure 5—figure supplement 1C***). *=nonspecific cellular protein. (**B**) Structure of NP (PDB: 2IQH) showing spatial organization of the PKCδ target sites identified by mass spectrometry. (**C**) Relative quantification of NP phosphorylation. Elution profiles of phosphopeptides identified by mass spectrometry were integrated to calculate the relative abundance of phosphorylation sites present in the groove (S165), tail loop (S402, S403, S407, S413), and body (T378) regions of the influenza nucleoprotein. (**D–F**) Loss of PKCδ impairs viral gene expression and replication. (**D**) Viral gene expression was measured in WT or PKCδ-knockout A549 cells infected with the indicated influenza reporter viruses or a West Nile virus (WNV) replicon. (**E**) Single-cycle virus replication measured 8 hpi in WT or PKCδ-knockout A549 cells infected with WSN virus. (**F**) Multi-cycle replication was measured in sgRNA control A549 cells or PKCδ-knockout cells. For all infections, data are shown as mean of n=3 ± standard deviation. *=p< 0.05 one-way ANOVA when compared to WT control. ns = not significant.

DOI: https://doi.org/10.7554/eLife.26910.012

The following figure supplements are available for figure 5:

**Figure supplement 1.** Identification of *PRKCD*−/− cells by Indel Detection by Amplicon Analysis (IDAA).
DOI: https://doi.org/10.7554/eLife.26910.013

**Figure supplement 2.** PKCδ is not required for entry via VSV-G or nucleo-cytoplasmic shuttling of NP.
DOI: https://doi.org/10.7554/eLife.26910.014

**Figure supplement 3.** Influenza virus replication is unchanged in clonal lines encoding a non-targeting CRISPR-Cas system.
DOI: https://doi.org/10.7554/eLife.26910.015

genome replication independent of its role in influenza virus entry (***Figure 5D***). Chemical inhibitor experiments have also suggested that PKCs play a role in the nucleo-cytoplasmic shuttling and localization of NP (***Bui et al., 2002***; ***Neumann et al., 1997***). Nonetheless, immunofluorescence assays showed that the kinetics of NP nuclear import and subsequent export to the cytoplasm were unchanged in infected *PRKCD*−/− cells, although total NP levels might be reduced (***Figure 5—figure supplement 2B***).

We next tested whether PKCδ is important for production of infectious progeny. Compared to WT cells, production of normal influenza virus in the knockout cell lines was reduced by up to 1000 fold in a single-cycle infection, consistent with the decreases in viral gene expression observed with our reporter viruses (***Figure 5E***). We created clonal A549 cell lines expressing a non-targeting control sgRNA to control for any effects that constitutive Cas9 expression might have on virus

replication. Viral replication in these cells was indistinguishable from the pooled parental A549 cells (*Figure 5—figure supplement 3*). However, viral titers were reduced by over 1.5 logs in both *PRKCD*$^{-/-}$ cell lines compared to the sgRNA control line in a multi-cycle replication assay (*Figure 5F*). Together, these data establish that phosphorylation of NP by endogenous PKCδ is crucial for viral gene expression, production of infectious progeny and multi-cycle replication. These reductions in replication likely represent combined effects of PKCδ knockout on both viral gene expression and entry (see below). As viral gene expression and replication was not completely abolished in the absence of PKCδ, they also implicate additional PKC family members or other host kinases as playing secondary or redundant roles in regulating NP phosphorylation.

## PKCδ regulates viral genome replication

Loss of PKCδ dramatically reduced NP phosphorylation and impaired viral gene expression (*Figure 5*). After entry, initial rounds of primary transcription occur on pre-formed viral RNPs deposited by the incoming virion and are dependent on attachment, entry and nuclear import. At later time points, the virus transitions to genome replication and secondary transcription in a process that requires NP oligomerization and formation of new RNPs. We probed the early steps during infection to identify the precise events impacted by PKCδ. Attachment and entry were measured with bioluminescent viral particles created with a reporter virus (PASN) that package the luciferase reporter into virions, allowing investigation of entry steps independent of the downstream events of vRNP nuclear import and transcription (*Tran et al., 2015*). Virions were bound to cells 4°C, subsequently shifted to 37°C to synchronize entry, and cells were then treated with cyclohexamide to ensure any detected reporter activity was derived only from incoming virions. Knockout of PKCδ reduced virion attachment or uptake ~2 fold (*Figure 6A*). To determine if this defect arose from attachment or fusion, a classic 'acid bypass' assay was performed where the viral membrane is fused to the plasma membrane by a transient low pH treatment causing vRNPs to be deposited directly into the cytoplasm (*Banerjee et al., 2014*; *Matlin et al., 1981*). Attachment and entry were indistinguishable for all cell lines in the acid bypass assay (*Figure 6A*), indicating that the lack of PKCδ does not alter attachment or pH-dependent fusion, but rather endosome-mediated uptake. Entry assay results agreed with prior reports showing that PKC inhibitors and mutants interfere with entry and trap influenza virions in the late endosome, suggesting that PKCδ is one of the isoforms involved in these early steps during infection (*Root et al., 2000*; *Sieczkarski et al., 2003*).

Our polymerase activity assays (*Figure 1*), RNP assembly experiments (*Figure 3E–F*), and infections with pseudotyped virus (*Figure 5D*) provided multiple lines of evidence that PKCδ is also important post-entry. To test this, we used the acid bypass approach to assess the role of PKCδ in viral gene replication independent of its role in entry. Infections were initiated via acid bypass with the PASTN reporter virus that requires viral gene expression for reporter activity (*Tran et al., 2013*). PKCδ knockout cell lines showed a significant reduction in viral gene expression at both 8 and 24 hr post-inoculation compared to parental A549 cells (*Figure 6B*). Thus, PKCδ directly impacts gene expression and possibly genome replication during infection.

Finally, we investigated how PKCδ regulates influenza virus RNA synthesis. Primary transcription is performed by pre-formed vRNPs deposited by the incoming virions. Transcripts from these vRNPs were quantified during infection by treating cells with cycloheximide to prevent the transition to genome replication and secondary transcription (*Barrett et al., 1979*). Cells were infected by acid bypass, treated with cycloheximide, and total RNA was then extracted and used in primer extension assays to measure incoming vRNPs (vRNA) and primary transcription (mRNA). vRNA and mRNA levels were similar in WT and *PRKCD*$^{-/-}$ cells, providing evidence that PKCδ is not required for nuclear import or primary transcription of incoming vRNPs (*Figure 6C*). This is consistent with the fact that NP is already oligomerized in these incoming vRNPs and need not be further regulated. In stark contrast, we detected significant reductions in replication and transcription when infections were allowed to proceed in *PRKCD*$^{-/-}$ cells (*Figure 6D*). cRNA levels were decreased by over 80% in knockout cells compared to the parental, indicating dramatic defects in genome replication. vRNA and mRNA levels were reduced by about 50%. The contribution of products from incoming RNPs, which do not require PKCδ, might possibly masking the full extent of the defects observed in replication and transcription. Thus, PKCδ plays an important role enabling genome replication and secondary transcription, events that both require formation of new RNPs.

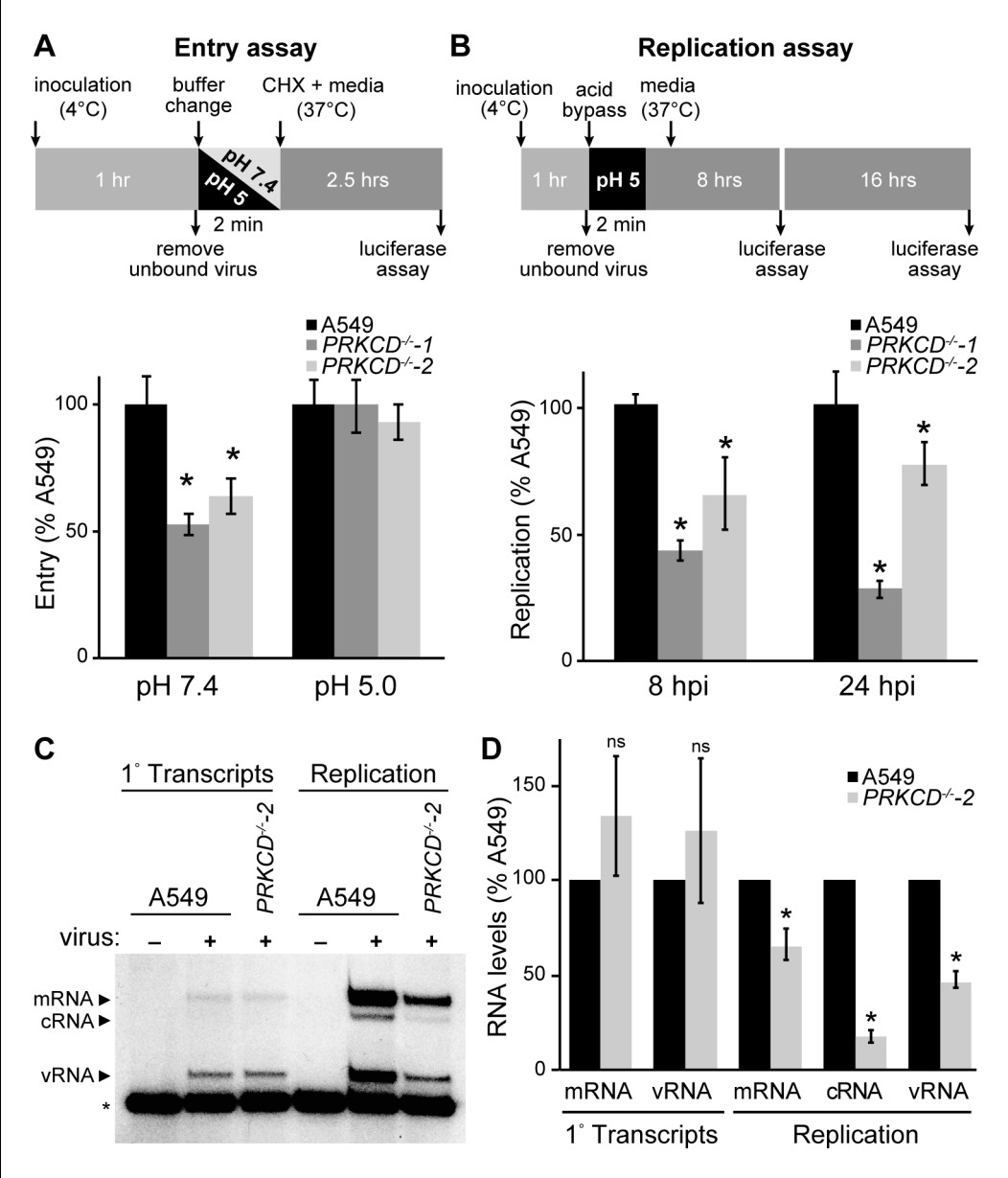

**Figure 6.** Loss of PKCδ specifically inhibits the transition to genome replication. (**A**) PKCδ plays a role in entry. The experimental diagram shows attachment of bioluminescent influenza virions (PASN) to cells at 4°C followed by normal receptor-mediated endocytosis (pH 7.4) or 'acid bypass' at the plasma membrane by exposure to low pH buffer (pH 5). Following entry, cells were treated with cycloheximide for 2.5 hr and entry was quantified. Data were normalized to WT cells for each condition. (**B**) To eliminate the effects of PKCδ on viral entry, infections with the transcription-dependent reporter virus (PASTN) were initiated by acid bypass as diagramed. Replication was quantified at 8 and 24 hpi. Data were normalized to WT cells for each condition. (**C**) and (**D**) Loss of PKCδ impairs replication but not primary transcription. (**C**) Primary transcription from incoming RNPs and replication by *de novo*-assembled RNPs were measured by primer extension assays in cells infected via acid bypass. To measure only primary transcription, cells were treated with cycloheximide (left). A non-specific product (*) served as a loading control. (**D**) Quantification of mRNA, cRNA and vRNA bands detected during primary transcription or replication in three independent primer extension experiments, normalized to WT A549 cells. For all experiments, data are mean of n=3± standard deviation. *p<0.05 one-way ANOVA when compared to WT A549 cells. ns = not significant.

DOI: https://doi.org/10.7554/eLife.26910.016

## Discussion

Influenza virus infections begin with a pioneering round of transcription from RNPs deposited by the incoming virion and transitions to genome replication and additional transcription at later time points. The polymerase copies genomic RNA while NP concomitantly oligomerizes along the length of the nascent product. NP changes during this process from an RNA-free monomer to a high-order RNA-bound oligomer in a process regulated by phosphorylation (*Mondal et al., 2015*; *Turrell et al., 2015*). Here we identify PKCδ as an important host regulator of RNP assembly and the resultant transition from transcription to replication. We show that PKCδ is recruited by PB2 to regulate NP oligomerization by phosphorylating both sides of the homotypic interface, targeting key phospho-sites in the binding grove (S165) and tail loop (S407) regions that mediate NP:NP interactions. Infections in PKCδ-knockout cells exhibited discrete defects in genome replication, a process that specifically requires NP oligomerization and RNP assembly, but not primary transcription, which is templated by pre-formed RNPs. This resulted in an overall reduction in viral gene expression and replication. Both hyper-phosphorylation by PKC over-expression or hypo-phosphorylation by PKCδ knockout or NP mutation disrupt the balance of NP oligomerization (*Mondal et al., 2015*), leading us to propose that dynamic phosphorylation establishes a localized pool of monomeric NP that is essential for forming nascent RNPs and the transition to genome replication (*Figure 7*). This raises the possibility that additional factors, possibly cellular phosphatases, may be required to license NP oligomerization and initiate genome replication. Protein phosphatase 6 was reported to directly bind multiple subunits of the viral polymerase and regulate RNA synthesis during infection, although it is not known if it targets any of these proteins for dephosphorylation (*York et al., 2014*). It is also possible that PKCδ activity or target specificity changes throughout infection to impact NP oligomerization potential. Viral product have also been proposed to regulate this transition, including NEP, svRNAs and *trans*-activating polymerases (*Jorba et al., 2009*; *Perez et al., 2010*; *Robb et al., 2009*; *York et al., 2013*). Thus, influenza virus deploys multiple approaches to ensure temporally regulated transcription and replication of the viral genome.

Influenza virus infection triggers multiple kinase cascades including phosphoinositide 3 kinase (PI3K), the Raf/MEK/ERK pathway, c-Jun N-terminal kinases (JNK) and PKCs (*Planz, 2013*). These kinases and their associated signaling pathways frequently play fundamental roles in mounting an

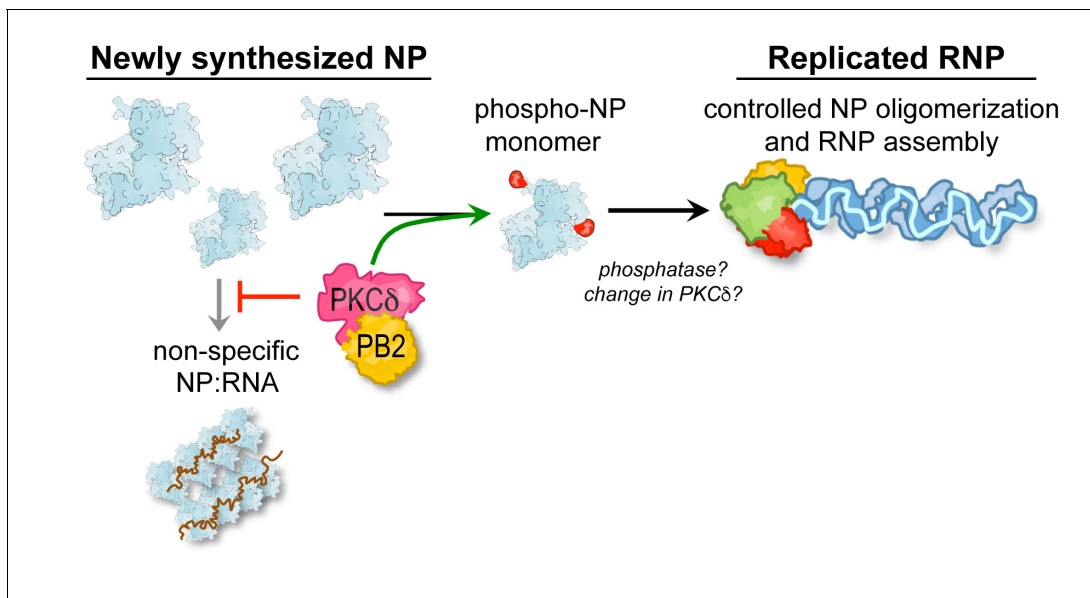

**Figure 7.** A model for PKC phospho-regulation of NP oligomerization and RNP assembly. PKCδ binds to the polymerase subunit PB2 to phosphorylate newly synthesized NP and negatively regulate its homo-oligomerization. This process is proposed to establish a pool of monomeric NP that is competent to specifically assemble with genomic RNA into viral RNPs, while also reducing premature non-specific assembly of NP:RNA aggregates. Our model further raises the possibility that additional factors ultimately license NP oligomerization and help potentiate successful genome replication.
DOI: https://doi.org/10.7554/eLife.26910.017

efficient antiviral response by the host cell. However, these same kinases are also exploited by the virus for efficient replication by facilitating viral entry, nuclear import of viral proteins, and the nuclear export of vRNPs (*Planz, 2013*). We show here that PKCδ is specifically required for regulating RNP assembly and the resultant replication of the viral genome. Another family member, PKCα, has also been shown to phosphorylate PB1-F2 at sites that are important for viral replication (*Mitzner et al., 2009*). *In vitro* assays have suggested that PKCα phosphorylates PB1 and NS1 (*Mahmoudian et al., 2009*), although our cell-based assays did not demonstrate any significant regulatory role for PKCα on polymerase function. Whereas these specific activities are associated with modifying viral proteins, PKCs play a more general role in viral entry by modifying cellular processes. Genetic ablation of PKCδ or PKCβ2 activity reduced viral entry, possibly by perturbing endosomal sorting and maturation (*Figure 6* and (*Sieczkarski et al., 2003*)). The involvement of multiple PKC isoforms at multiple stages during infection suggests that broad-spectrum PKC inhibitors may be especially effective anti-viral compounds, even at concentrations that do not fully block PKC activity or their normal cellular functions. This may extend beyond influenza virus, as PKCs have been shown to regulate gene expression and replication of other negative-strand RNA viruses by phosphorylating their P proteins, including human parainfluneza virus 3, Sendai virus, and rabies virus (*De et al., 1995*; *Gupta et al., 2000*; *Huntley et al., 1997*).

NP contains multiple serine and threonine phosphorylation sites throughout the length of the protein (*Hutchinson et al., 2012*; *Mondal et al., 2015*). Our data show that sites targeted by PKC during infection reside primarily at the tail loop:groove interface (*Figure 5*). S165 in the NP groove lies within a linear consensus recognition motif for PKCs, RxxS, where arginine at the −3 position facilitates kinase-substrate interactions (*Nishikawa et al., 1997*). Both NP R162 and S165 in the PKC recognition motif are highly conserved in influenza A viruses. NP S403 also lies within a PKC recognition motif, although the phospho-site and recognition motif are not well conserved. In contrast, other residues including S402, S407 and S413 do not reside within linear motifs and may rely on structure-based recognition motifs formed by non-contiguous regions of the protein (*Duarte et al., 2014*). While kinase recognition motifs help to establish a specific kinase-substrate interaction, PKC and PKA family members often rely upon additional co-factors to phosphorylate target proteins (*Mochly-Rosen and Gordon, 1998*; *Welch et al., 2010*). PKCs pair with anchoring proteins, collectively termed receptor of activated C-kinase (RACKs), to enhance kinase activity and stability, increase their affinity and selectivity towards specific substrates, and integrate multiple signaling pathways. Our data suggest that PB2 performs an analogous function anchoring PKCδ and NP. Like RACKs, PB2 anchors PKCδ and NP through non-overlapping protein-protein interactions; PB2 helps to convey specificity, preferentially interacting with full-length PKCδ compared to full-length PKCβ2 or PKCη; and PB2 anchors activated PKCδ as evidenced by kinase activity in *in vitro* reactions without the need for exogenous activators (*Figures 3–4*). It is tempting to speculate that binding to PB2 might activate or stabilize the activated conformation of PKCδ, as has been shown for other RACK: PKC pairings (*Mochly-Rosen and Gordon, 1998*), or provide increased recognition of the non-canonical site at S407. It will be important to determine the sites of interaction between PB2 and PKCδ, how this conveys preferential interaction with only one of the eleven different PKC isoforms, and if these interactions affect the activation state of PKC.

While PKCδ is a major kinase that phospho-regulates NP oligomerization, the RACK-like activity of PB2 suggests that it is the NP:PB2:PKCδ complex that provides specificity and efficient modification of NP. It thus appears that the polymerase itself, or perhaps just the PB2 subunit alone, may contribute to the PKC-mediated phospho-regulation of RNP assembly. The PB2-phosphoNP complexes may function in *cis* to regulate the initial stages of RNP assembly or possibly in *trans* to shuttle monomeric NP to already growing RNPs, analogous to the role of the phosphoprotein P from non-segmented negative-sense RNA viruses that chaperones monomeric nucleocapsid proteins to replicating RNPs (*Masters and Banerjee, 1988*). It is not known how NP is recruited to the growing oligomer chain, but regardless of the route of recruitment, NP must ultimately be licensed for oligomerization to form an RNP. This adds another layer of control, and may involve altered PB2:PKC and PB2:NP interactions, changes to the activation state of PKCδ associated with PB2, or possibly the involvement of cellular phosphatases. As both over expression and knockout of PKCδ perturb RNP formation, it is clear that the activity of PKCs must be finely balanced to enable NP oligomerization when and where it is needed. In summary, we have shown that influenza virus exploits the host

kinase PKCδ to phospho-regulate NP oligomerization, revealing a complex regulatory pathway controlling RNP formation and the transition from transcription to replication during the viral life cycle.

## Materials and methods

### Plasmids, viruses, antibodies and cells

All virus related genes were derived from influenza A/WSN/33 virus. NP and polymerase proteins were expressed in cells from the plasmids pCDNA6.2-NP-V5, pCDNA3-PB2-FLAG (encoding a C-terminal FLAG tag) or pCDNA3-PA and pCDNA3-PB1. vNA-luc reporter plasmids encode firefly luciferase in the negative sense flanked by UTRs from the NA gene (*Regan et al., 2006*). pET28a-NΔ7NP was used for bacterial expression of NP with a C-terminal His tag as described previously (*Mondal et al., 2015*). Plasmids expressing full-length PKC isoforms (β2, δ, η) or just the catalytic domains (PKC-CAT α, β1, β2, γ, δ, ε, η, ι) were previously described (*Table 2*) (*Soh and Weinstein, 2003*) (Addgene plasmids #21234, 16380, 16384, 21238, 16388, 21242, 21247, 21254). PKCθ-CAT was prepared by cloning the catalytic domain from the full length isoform into pHACE. PKC-CAT domains were mutated to catalytically inactive forms replacing specific lysine residues in the catalytic domains to either arginine or to methionine following the approach used for full-length PKC (*Soh and Weinstein, 2003*).

Infections were performed with A/WSN/1933 (H1N1), WSN stably-encoding PB2 with a C-terminal FLAG tag (WSN-PB2-FLAG) (*Dos Santos Afonso et al., 2005*) or PA-2A-Swap-Nluc (PASTN) reporter viruses based on the strains A/WSN/33 (H1N1) and A/California/04/2009 (H1N1) (*Karlsson et al., 2015*; *Tran et al., 2015*; *2013*). Entry assays were performed with the WSN-based reporter virus PA-SWAP-NLuc (PASN) that packages Nanoluc into virions (*Tran et al., 2015*). The WSN-based reporter virus encoding VSV-G in place of HA and *Renilla* luciferase (FVG-R) or GFP (FVG-G) in place of NA was used as described (*Hao et al., 2008*; *Watanabe et al., 2003*). The West Nile virus replicon, where structural genes were replaced with *Renilla* luciferase, was prepared as described (*Lo et al., 2003*; *Shi et al., 2002*).

Antibodies used include: anti-PKCδ (sc-937 C20, Santa Cruz Biotech), anti-HA clone 3F10 (Roche), anti-V5 (R961-25, Invitrogen), anti-GFP (B-2, Santa Cruz Biotech), anti-NP (H16-L10-4R5) (*Yewdell et al., 1981*), anti-FLAG M2 (Sigma), anti-PB1 and anti-PB2 (*Mehle and Doudna, 2008*), and anti-influenza virus RNP (BEI Resources NR-3133).

293T (CRL-3216), MDCK (CCL-34), and A549 (CCL-185) cells were purchased as authenticated stocks from ATCC. All cells were maintained in Dulbecco's modified Eagle's medium (DMEM) supplemented with 10% FBS at 37 °C and 5% $CO_2$. Cells are tested for mycoplasma contamination approximately once a month using MycoAlert (Lonza LT07-218).

**Table 2.** Domain boundaries and mutations for PKC variants.

| PKC isoform | CAT domain (aa) | DN mutation |
| --- | --- | --- |
| PKCα | 326–672 | K368R |
| PKCβ1 | 329–671 | K371R |
| PKCβ2 | 329–673 | K371R |
| PKCδ | 338–697 | K380R |
| PKCε | 334–674 | K376R |
| PKCη | 342–683 | K384R |
| PKCι | 239–592 | K281M |
| PKCζ | 232–587 | K273M |

Boundaries and mutations for PKC expression constructs: PKC catalytic fragments (CAT) are defined by the first and last amino acid residues in the expression construct. Dominant negative (DN) mutations were created by introducing the indicated mutation.

DOI: https://doi.org/10.7554/eLife.26910.018

## Polymerase activity assay

Activity assays were performed following our prior approach by transfecting 293T cells in triplicate with plasmids expressing PA, PB1, PB2, NP and negative-sense vNA-luciferase reporter (*Kirui et al., 2016*) (*Figure 1—figure supplement 1*). Where indicated, cells were co-transfected with plasmids expressing PKC catalytic domains or dominant negative versions. Polymerase activity was measured using the luciferase assay system (Promega) and NP expression was confirmed by western blotting.

## Immunoprecipitations

293T cells expressing HA-tagged PKC, V5-tagged NP and polymerase proteins were lysed in radio-immunoprecipitation assay (RIPA) buffer (50 mM Tris-HCl [pH 7.5], 150 mM NaCl, 2 mMEDTA, 1% NP-40, 0.5% deoxycholate, 0.1% SDS) supplemented with 5 mg/ml of BSA and clarified by centrifugation. Lysates were incubated with anti-HA antibody and immunocomplexes were captured on Protein A Dynabeads (Invitrogen), washed extensively and analyzed by western blotting.

## Primer extension analysis

A549 cells were infected with WSN using the low pH fusion buffer as described above at an MOI of 10 to measure primary transcription or at an MOI of 1 to measure replication. After attachment and low pH-mediated entry, cells were either incubated in VGM for 8 hr to measure replication or 6 hr with 1 mM cyclohexamide to measure primary transcription or incubated. Total RNA was isolated using TRIzol reagent (Invitrogen) and primer extensions assays were performed as described using primers using primers to detect viral NA RNAs and host 5S RNA (*Mehle and Doudna, 2008*; *Robb et al., 2009*). Transcription products were resolved via UREA-PAGE, quantified by phosphorimaging, and analyzed using ImageQuant TL software (GE Healthcare).

## Immunofluorescence assays

Wild type or PKCδ knockout cells grown on coverslips were infected with WSN at an MOI of 2. Virion binding was performed at 4 °C for 1 hr and synchronous infections were initiated by shifting cells to 37 °C. Cells were fixed at different time points post-inoculation with 3% formaldehyde and permeabilized with 0.1M Glycine/0.1% Triton-X 100 in PBS for 10 min at room temperature. After blocking with 4 °C with 3% BSA overnight, NP was detected with anti-RNP antibody and Alexa Fluor 488-conjugated donkey anti-goat IgG (Cell Signaling). Cells were imaged using Zeiss Axio Imager M2 and post-processed with ImageJ.

## *In vitro* kinase assay

The *in vitro* kinase assay was adapted from the protocol originally described by Soh *et al.* (*Soh and Weinstein, 2003*). Briefly, 293 T cells were transfected with PKC-CAT or control plasmids, cells were lysed in PKC extraction buffer (50 mM HEPES [pH 7.5], 150 mM NaCl, 0.1% Tween-20, 1 mM EDTA, 2.5 mM EGTA, 10% glycerol, protease and phosphatase inhibitors), and PKCs were immunoprecipitated with anti-HA antibody. Immune complexes were captured with Protein A Dynabeads, washed in extraction buffer, washed in PKC reaction buffer (50 mM HEPES [pH 7.5], 10 mM MgCl2, 1 mM dithiothreitol, 2.5 mM EGTA), and finally resuspended in 20 µl of the reaction buffer to provide the source of kinase used in each assay. Wild type or mutant recombinant NP was purified from bacteria following our existing protocol and treated with RNaseA for 2 hr prior to use as substrate (*Mondal et al., 2015*). 2 µg of NP was reacted with equivalent amount of kinase or control complexes in the presence of 10 µCi of γ-$^{32}$P ATP at 37°C for 1 hr. Reactions were terminated by boiling in SDS sample buffer and analyzed by SDS-PAGE gel electrophoresis and autoradiography. Where mentioned, the PKC inhibitor 1-(5-isoquinolinesulfonyl)−2-methylpiperazine (H7) (Sigma) was added to the reaction.

For *in vitro* kinase assays with endogenous PKCδ, A549 cells were infected with WSN-PB2-FLAG virus at an MOI of 5 and harvested at 6 hr of post-inoculation. Cells were lysed in PKC extraction buffer and PB2-FLAG was immunoprecipitated with M2 affinity resin (Sigma). Immuno-captured complexes were resuspended in PKC reaction buffer and used as a source of both kinase and substrate. Samples were further treated with RNaseA, RNaseA and H7, or left untreated prior to performing a kinase reaction as described above.

## Virus replication and entry assays

Multicycle replication assays were performed by infecting A549 cells with WSN at an MOI of 0.001. Aliquots were collected at the indicated time points and viral titers were measured by plaque assay on MDCK cells.

Entry assays were performed by incubating WT or *PRKCD*$^{-/-}$ A549 cells with PASN at an MOI of 5 in virus growth media (VGM: DMEM, 0.2% bovine serum albumin (BSA), 25 mM HEPES buffer, and 0.25 µg/ml TPCK-trypsin) at 4°C for 1 hr followed by washing with cold PBS to remove unbound virus particles. Acid bypass assays were modified from previous approaches (*Banerjee et al., 2014*; *Matlin et al., 1981*). Following virion binding, cells were incubated in low pH fusion buffer (50 mM citrate, pH 5.0, 154 mM NaCl) or mock-treated with pH 7 buffer (20 mM HEPES, pH7.4, 154 mM NaCl) for 2 min at 37°C and washed with cold PBS. Finally, pre-warmed VGM supplemented with 1 mM cycloheximide was added and the cells were incubated at 37°C for 2.5 hr. Virion entry was quantified using the Nano-Glo luciferase assay (Promega).

Gene expression by West Nile virus replicons was measured by inoculating *WT or PRKCD-/-* A549 cells at an MOI of 0.1. *Renilla* luciferase activity was measured 24 hr post-inoculation. PASTN reporter virus replication assays were initiated by acid bypass at an MOI of 0.05. Cells were incubated at 37°C for 8 and 24 hr and viral gene expression was measured using the Nano-Glo luciferase assay (Promega).

## PKCδ knockout cell line preparation

sgRNAs targeting the fourth exon of *PRKCD* (aacgatgaaccgccgcggag) or a non-targeting control (ACGGAGGCTAAGCGTCGCAA) were cloned into lentiCRISPR v2 (Addgene plasmid # 52961, Feng Zhang (*Sanjana et al., 2014*)). Lentiviral particles were produced with the resulting plasmids and used to transduce A549 cells. Transduced cells were selected with puromycin and single-cell sorted by FACS. Following outgrowth, clonal lines were screened for homozygous *PRKCD* gene disruptions by Indel Detection by Amplicon Analysis (IDAA) using primers that amplify the edited locus to identify genomic deletions of the target locus (*Yang et al., 2015*) (*Figure 5—figure supplement 1A*). Gene disruption was confirmed by Sanger sequencing of IDAA amplicons and identified the clonal lines *PRKCD*$^{-/-}$−1 and *PRKCD*$^{-/-}$−2 containing homozygous non-sense mutations (*Figure 5—figure supplement 1B*). Knockout was confirmed by western blotting of cell lysate. All assays were performed with early passages of knockout cell lines.

## Mass spectrometry

WT or *PRKCD*$^{-/-}$−1 and *PRKCD*$^{-/-}$−2 A549 cells were infected with WSN at an MOI of 5. Cells were harvested at 4 and 8 hpi, pooled, lysed and NP was purified by immunoprecipitation as described previously (*Mondal et al., 2015*). Purified protein samples were lyophilized, dissolved in 8 M urea, reduced and alkylated with 10 mM tris(2-carboxyethyl)phosphine and 40 mM chloroacetamide, diluted to a final concentration of 1.5 M urea using 50 mM Tris, and digested with trypsin overnight at room temperature. Resultant peptides were desalted using a C18 Strata X column (Phenomenex) and enriched for phosphorylation by immobilized metal affinity chromatography using Ni-NTA magnetic agarose beads (Qiagen) (*Rose et al., 2012*). Both non-phosphorylated and phosphorylated peptide samples were resuspended in 0.2% formic acid and analyzed by MS. A 100 min nano-liquid chromatography gradient was used to introduce peptides to an Oribtrap Elite mass spectrometer (Thermo Scientific) and peptides were analyzed by data dependent acquisition (DDA) using higher-energy collisional dissociation (HCD) to fragment them (*Vincent et al., 2013*). Spectra obtained from these DDA experiments were searched against a concatenated target-decoy database containing the protein sequences of Homo sapiens and influenza A virus (Uniprot) using both Sequest within the Proteome Discoverer software package (Thermo Fisher) and MaxQuant (*Cox et al., 2011*; *Cox and Mann, 2008*). For all samples, cysteine carbamidomethylation and methionine oxidation were searched as fixed and variable amino acid modifications, respectively, and phosphorylation of serine, threonine, and tyrosine residues were searched as variable modifications. Database searches utilized a precursor mass tolerance of 40 ppm and a fragment ion tolerance of 0.02 Da, with peptide identifications filtered to a 1% false discovery rate (FDR). Proteome Discoverer searches used PhosphoRS to localize phosphorylation to amino acid residues using a fragment mass tolerance of 0.02 Da, automatically considering neutral loss peaks for HCD and a maximum of 500

position isoforms per phosphopeptide (*Taus et al., 2011*). Phosphosites were determined as localized if they were scored with a localization probability >75%. Influenza NP phosphopeptides were identified from DDA MS runs, including one peptide with phosphorylation localized to the NP T378. Subsequent targeted MS experiments localized phosphorylation to the S165 site on the tryptic NP peptide MCSLMQGSTLPR. These targeted MS experiments also monitored the *m/z* values for this peptide with one or two oxidized methionine residues. Four phosphopeptide isoforms for the NP peptide ASSGQISIQPTFSVQR were targeted for relative quantification. Using HCD fragmentation, phosphorylation was localized to the S402, S403, S407, and S413 residues on this peptide (*Mondal et al., 2015*). Extracted ion chromatograms were generated for the six NP phosphopeptides modified at S165, S402, S403, S407, S413, and T378. The peak area for each peptide was normalized to the total NP protein loaded on-column for each lysate, and the relative abundance of each phosphopeptide was compared between the lysates of infected wild-type A549 or mutant $PRKCD^{-/-}$ KO cell lines.

## Statistics

Data are representative of at least three independent experiments, with each experiment performed in triplicate or greater. Multiple comparisons were performed with a one-way ANOVA and statistical significance was indicated when $p < 0.05$.

## Acknowledgements

This work was supported by the National Institute of General Medical Sciences (R00GM088484), National Institute of Allergy and Infectious Diseases (R01AI125271) and a Shaw scientist award to AnM, the National Institute of Allergy and Infectious Diseases (T32AI078985) to ARD, and the National Institute of General Medical Sciences (R35GM118110) to JJC. AnM holds an Investigators in the Pathogenesis of Infectious Disease Award from the Burroughs Wellcome Fund. Anti-influenza Virus RNP antiserum (NR-3133) was obtained through BEI Resources, NIAID, NIH. We thank B Weinstein and F Zheng for reagents deposited in Addgene, D Rubenstein for technical advice, VG Tran for help with diagrams, and M Harrison and members of the Mehle lab for critical reading of the manuscript.

## Additional information

### Funding

| Funder | Grant reference number | Author |
|---|---|---|
| Greater Milwaukee Foundation | Shaw Scientist Award | Arindam Mondal<br>Anthony R Dawson<br>Elyse C Freiberger<br>Steven F Baker<br>Joshua J Coon<br>Andrew Mehle |
| American Lung Association | RG-310016 | Arindam Mondal<br>Andrew Mehle |
| National Institute of Allergy and Infectious Diseases | R01AI125271 | Arindam Mondal<br>Anthony R Dawson<br>Elyse C Freiberger<br>Joshua J Coon<br>Andrew Mehle |
| National Institute of General Medical Sciences | R35GM118110 | Gregory K Potts<br>Joshua J Coon |
| National Institute of General Medical Sciences | R00GM088484 | Andrew Mehle |
| National Institute of Allergy and Infectious Diseases | T32AI078985 | Andrew Mehle |

| Burroughs Wellcome Fund | Investigators in the Pathogenesis of Infectious Disease | Andrew Mehle |
|---|---|---|

The funders had no role in study design, data collection and interpretation, or the decision to submit the work for publication.

## Author contributions

Arindam Mondal, Conceptualization, Formal analysis, Investigation, Methodology, Writing—original draft, Writing—review and editing; Anthony R Dawson, Formal analysis, Investigation, Methodology, Writing—review and editing; Gregory K Potts, Formal analysis, Investigation, Methodology, Writing—original draft, Writing—review and editing; Elyse C Freiberger, Formal analysis, Investigation, Methodology; Steven F Baker, Conceptualization, Investigation, Methodology; Lindsey A Moser, Conceptualization, Formal analysis, Investigation, Methodology; Kristen A Bernard, Conceptualization, Formal analysis, Supervision, Project administration; Joshua J Coon, Andrew Mehle, Conceptualization, Formal analysis, Supervision, Funding acquisition, Writing—original draft, Project administration, Writing—review and editing

## Author ORCIDs

Andrew Mehle http://orcid.org/0000-0001-6060-4330

## Decision letter and Author response

Decision letter https://doi.org/10.7554/eLife.26910.019
Author response https://doi.org/10.7554/eLife.26910.020

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
