## [Decision Letter]

[Editors’ note: a previous version of this study was rejected after peer review, but the authors submitted for reconsideration. The first decision letter after peer review is shown below.]

Thank you for submitting your work entitled "Influenza virus recruits PKCδ to regulate the transition from transcription to replication" for consideration by *eLife*. Your article has been reviewed by two peer reviewers, and the evaluation has been overseen by a Reviewing Editor and Wenhui Li as the Senior Editor. The reviewers have opted to remain anonymous.

Our decision has been reached after consultation between the reviewers. Based on these discussions and the individual reviews below, we regret to inform you that your work will not be considered further for publication in *eLife*.

*Reviewer #1:*

In this interesting manuscript from Mondal et al., the authors present evidence that PKCδ influences influenza A virus replication in mammalian cells in culture. To form this conclusion the authors use a variety of approaches including a minigenome assay, overexpression of isoforms of PKC and catalytic mutants of PKC and engineer cells to lack specific PKC isoforms in which they study influenza replication. Those approaches are combined with mass-spectrometry to study sites of PKC modification of the viral NP protein, and in vitro phosphorylation by PKC. The data presented are comprehensive for a cell culture analysis and save for some issues regarding variation in NP levels apparent from the westerns, the data are consistent with a role of PKC in phosphorylation of NP. The data demonstrate that influenza virus NP can be phosphorylated by PKC, that association of PKC and NP seems to be bridged by the polymerase subunit PB2 and that phosphorylation regulates NP oligomerization. This set of data leads the authors to conclude that PKC regulates the transition from transcription to replication.

Understanding how host factors are coopted by viruses to aid in their replication cycle is intrinsically biologically interesting, although it is unclear whether this could be therapeutically useful. Although this is an important piece of work, there are some areas that would benefit from following issues addressed.

1) My most significant comment relates to the magnitude of the effect of loss of PKCδ on polymerase activity and viral replication. If PKC is playing a critical role in regulating the NP oligomerization I am surprised by the relative low impact on replication in A549 cells as judged by the reporter assay (Figure 6) and even the primer extension assay (Figure 6). Although 6D shows a reduction in cRNA levels the vRNA and mRNA levels are only suppressed about 50%. The process of replication is exponential, and by suppressing a "key" step of that process this reviewer was rather surprised to see a fairly modest impact on the steady state levels of RNA accumulated over a 24h time period.

2) The title of the paper is an over interpretation. What the authors present is data that PKC can influence the oligomerization of NP. While NP is required for replication it is not clear that PKCδ directly regulates the transition from transcription to replication. Rather it regulates a step that is important for replication.

3) The step that appears to be regulated is the oligomerization of NP from a pool of presumably soluble precursors in the cell. If nascent chain assembly with NP is regulated by PKCδ then that process should be readily visible by a pulse chase experiment using a pool of labeled NP. Such an experiment would be quantitative and would read out directly on the very step the authors claim is regulated by PKC.

4) The quantitative nature of the Western blots is uncertain. There is considerable variation in the level of NP that is detected (Figure 1, Figure 2). The variation is important since there seems to be a correlation in the amount of plasmid expressed NP that correlates with the lower polymerase activities (see Figure 1 lower polymerase activity and active overexpressed PKC isoforms and Figure 1). Although some of the NP is in a phosphorylated form the total amount of NP should be carefully quantified since this could lead to considerable variation in the luciferase assays.

*Reviewer #2:*

Reversible phosphorylation of NP establishes a pool of RNA-free NP that can be recruited into growing vRNPs. Since NP phosphorylation and oligomerization is regulated by PKC the authors designed experiments to identify the responsible PKC isoform, namely PKCδ. The authors suggest that PB2 recruits this PKC isoform to phosphorylate NP. Finally knock-out of PKCδ result in impaired viral replication due to deficits in viral uptake/uncoating and replication. Based on these results the authors conclude that influenza virus recruits PKCδ to regulate the transition from transcription to replication.

The paper is well written and the conclusions are partially supported by the obtained data. Following open issues might be addressed:

1) The authors show that PKCδ phosphorylates NP to keep it in the monomeric form. As a result this protein is not available for the synthesis of full-length vRNA and cRNA. The transition from transcription to replication must be therefore regulated by an unknown phosphatase and not by PKCδ.

2) The authors speculate that PB2 bridges the binding of PKCδ to NP, however, a direct interaction between PKCδ and PB2 was not shown. Interestingly, PB2 itself seems to be phosphorylated by PKCδ based on the appearance of a higher migrating band (Figure 2). If true, PB2 might be a substrate of PKCδ and therefore co-purifies with this PKC isoform.

3) Phosphorylation of NP occurs independent on the presence of PB2 (Figure 2). Thus it is difficult to understand why PB2 should be important to bridge the interaction of PKCδ with NP.

4) The extend of NP phosphorylation in the presence of full length PKCδ seems to be negligible (e.g. Figure 2: no slower migrating form of NP in the Input lane). How does this translate to the inhibitory effect exerted by these kinases?

5) Figure 3. The headline of the figure legend should be rephrased, since no experimental evidence was provided that PKC regulates RNP assembly.

6) Figure 4: Without further purification of the vRNPs, it is unclear whether RNP-bound NP is analyzed or NP purified with PB2.

7) Figure 6: There is a substantial effect on entry in both the *PRKCD^-/-^*-1 and -2 cells (panel A, which might significantly affect viral growth efficiency. Moreover, replication seems to be affected too at early time points post infection. It is however unclear which block might contribute most to the observed growth deficits.

8) Figure 6: Interference was also measured in the so-called "Replication assay" which was initiated by an acid bypass. In this very artificial assay it is unclear how the vRNPs are actually transported to the cell nucleus and whether PKCδ affect this transport step. Results provided in panel C might suggest that primary transcription in the presence of CHX is not affected. Quantification of triplicates are required to draw this conclusion. In addition, primary transcription of viral mRNA should be determined after "normal" infection of wt and *PRKCD^-/-^*-1 or -2 cells in the presence of CHX.

[Editors’ note: what now follows is the decision letter after the authors submitted for further consideration.]

Thank you for resubmitting your work entitled "Influenza virus recruits host protein kinase C to control assembly and activity of its replication machinery" for further consideration at *eLife*. Your revised article has been favorably evaluated by Wenhui Li (Senior Editor), a Reviewing Editor and two reviewers.

The data provided do a fantastic job of thoroughly examining interactions between nucleoprotein and several PKC isoforms. The manuscript is also clearly written and relatively easy to follow. The manuscript has been greatly improved and has been well-received. Additional revisions are requested for purposes of interpretation and clarity.

Scientific comments:

1) While the presented data are very solid and the interpretations leave no obvious alternate explanations, the apparently opposite roles of PKC in early transcription and subsequent replication of the viral genomes, via phosphorylation of NP monomers and modulation of NP oligomerization need to be better discussed. While the authors point out some of these conundrums, a figure should be provided that models the phosphorylation/oligomerization of NP in transcription and replication.

2) Putative phosphatases that are postulated to be involved if PKC phosphorylates NP late in infection could be discussed. It would be interesting to comment on any evidence that PKC substrate specificity is dynamic during infection.

3) In general, a probe for the component actually subjected to the immunoprecipitation in the post IP fraction would be nice, otherwise it is hard to determine if the observed lack of enrichment (or surplus of enrichment) is due to differential immunoprecipitation or bone-fide differences in affinity. This becomes particularly a concern owing to different PKC isoforms serving as the target for IP rather than a single target with interaction partners varying. Presumably, those data have been obtained or can be obtained. Some reassurance that recovery of the targeted protein is consistent, and comparable between the iso forms of PKC, should be provided.

4) Figure 3. Is it known what other kinase target sites are observed in panel D (S165A/S470A)? Can this be discussed? Why does PKC2 not allow the oligomerization of GFP-TL and NPΔTL? Clearly, it can phosphorylate NPs.

5) Figure 4. Has the oligomerization status of NP and associated PB2 before and after the kinase assay? This would reveal whether RNP-associated NP can be phosphorylated. As a monomer or oligomer?

Comments about presentation:

1) In Figure 2, panels A, C, and D have bands that are clearly overexposed. While we understand the difficulty in displaying data with a wide dynamic range, including additional, less-exposed, panels would allow for better comparisons between samples. Panel D also has a far under-loaded immunoprecipitation control relative to the experimental samples.

2) The figures are dense and even though the text follows the data closely, additional clarity such as the addition of lane numbers would greatly improve the exposition.

3) Additional clarity would be provided if Figure 1 (i.e. Polymerase assays) and 3 (i.e. RNP assembly assays) had top diagrams as in Figure 6.

4) In the last paragraph of the subsection “PB2 stabilizes interactions between PKCδ and NP”, the NP:PB2:PKCδ complex is termed a heterotrimer. While this may be stoichiometrically correct with respect to these components, higher-order versions or other interaction partners could be involved. Demonstrating this is unnecessary for the present manuscript and a simple change in the language to hetero-oligomer would be more than sufficient.

---

## [Author Response]

[Editors’ note: the author responses to the first round of peer review follow.]

*Reviewer #1:*

*[…] Understanding how host factors are coopted by viruses to aid in their replication cycle is intrinsically biologically interesting, although it is unclear whether this could be therapeutically useful. Although this is an important piece of work, there are some areas that would benefit from following issues addressed.*

*1) My most significant comment relates to the magnitude of the effect of loss of PKCδ on polymerase activity and viral replication. If PKC is playing a critical role in regulating the NP oligomerization I am surprised by the relative low impact on replication in A549 cells as judged by the reporter assay (Figure 6) and even the primer extension assay (Figure 6). Although 6D shows a reduction in cRNA levels the vRNA and mRNA levels are only suppressed about 50%. The process of replication is exponential, and by suppressing a "key" step of that process this reviewer was rather surprised to see a fairly modest impact on the steady state levels of RNA accumulated over a 24h time period.*

We agree with this reviewer that if PKCδ was the sole kinase regulating NP oligomerization, then the effects we report would be underwhelming. However, our data demonstrate that this is not that case and we apologize if our initial submission over-interpreted the results to present PKCδ as a sole regulator. While PKCδ was the most prominent stable interactor with viral proteins (Figure 2), and thus the primary focus of our work, Figure 1–Figure 3 show that other PKCs interact with and phosphoregulate NP function. Notably, we show that PKCη also phosphorylates residues at the NP:NP interface and disrupts NP:NP interactions (Figure 3). Furthermore, quantitative mass spectrometry revealed that the key phospho-regulatory sites NP S165 and S407 are still phosphorylated in cells lacking PKCδ, although at a significantly reduced level (Figure 5). Thus, PKCδ is an important regulator, as revealed by defects in gene expression and viral replication, but there is also likely some degree of redundancy afforded by other PKC isoform (or possibly other kinase families). We now make this point explicit in the second paragraph of the subsection “NP phosphorylation and influenza virus replication are impaired in PKCδ-deficient cells”. Moreover, in the last paragraph of the subsection “PKCδ regulates viral genome replication”, we note that some of the apparent “modest” reduction detected in replication and primer extension assays could be due to the fact that incoming pre-formed RNPs, which do not rely on PKCδ, are still competent to express mRNA and might possibly mask the full extent of the defect.

We performed one more additional control experiment to ensure that these decreases in influenza gene expression and replication were significant and not simply generic defects in virus replication. We infected our WT and knockout cells with a West Nile virus replicon. There was no significant difference in WNV gene expression in the presence or absence of PKCδ, indicating that loss of PKCδ does not cause generic reductions in viral entry or gene expression

(Figure 5 and subsection “NP phosphorylation and influenza virus replication are impaired in PKCδ-deficient cells”, third paragraph). The results also afford at least some degree of specificity of PKCδ for influenza virus replication and reinforce that the decreases we detected were a biologically meaningful.

*2) The title of the paper is an over interpretation. What the authors present is data that PKC can influence the oligomerization of NP. While NP is required for replication it is not clear that PKCδ directly regulates the transition from transcription to replication. Rather it regulates a step that is important for replication.*

We agree with the reviewer and apologize for our lack of precision. We have since modified the title to more accurately reflect our data: Influenza Virus Recruits Host Protein Kinase C to Control Assembly and Activity of Its Replication Machinery.

*3) The step that appears to be regulated is the oligomerization of NP from a pool of presumably soluble precursors in the cell. If nascent chain assembly with NP is regulated by PKCδ then that process should be readily visible by a pulse chase experiment using a pool of labeled NP. Such an experiment would be quantitative and would read out directly on the very step the authors claim is regulated by PKC.*

The reviewers both noted that we had not formally demonstrated a defect in RNP assembly (both here and below in reviewer 2, point 5). We undertook experiments to directly assess RNP formation in the presence or absence of active PKCδ. RNP formation was measured by immunoprecipitating PA and testing for co-precipitation of NP. As PA and NP do not directly interact, any NP that co-precipitates is presumed to be part of an RNP. We show that NP coprecipitates only upon PA capture, and that the amount of NP is markedly reduced when PKCδ is present (Figure 3). These data also show that PKCδ expression does not interfere with polymerase trimer formation, another step required for RNP assembly. From these results, we conclude that PKCδ expression decreases NP oligomerization and incorporation into RNPs and reduces RNP steady-state levels (subsection “PKC phosphorylates NP at the tail loop:groove interface and blocks oligomerization”, last paragraph).

The reviewer had suggested pulse-chase experiments to measure the rate of NP incorporation. That is also a very interesting experiment, although it addresses a slightly different question. We performed initial pilot experiments and subsequent optimization experiments. We could detect polymerase trimer formation and RNP assembly, but there were a number of confounding factors. Perhaps the biggest was that if the experiments were performed on infected cells with or without PKCδ, the levels of total NP were different as viral gene expression is affected by PKCδ (Figure 1, Figure 5, Figure 6). As the rate of incorporation is likely dependent on the concentration of NP, this approach could not isolate the effects of PKCδ on RNP assembly rates from its effects on gene expression. Instead, we opted for the more straightforward experiment described above that controlled for protein expression levels with or without PKCδ and demonstrated that PKCδ decreased the level of RNPs.

*4) The quantitative nature of the Western blots is uncertain. There is considerable variation in the level of NP that is detected (Figure 1, Figure 2). The variation is important since there seems to be a correlation in the amount of plasmid expressed NP that correlates with the lower polymerase activities (see Figure 1 lower polymerase activity and active overexpressed PKC isoforms and Figure 1). Although some of the NP is in a phosphorylated form the total amount of NP should be carefully quantified since this could lead to considerable variation in the luciferase assays.*

This is always a concern of ours too, and why we performed all of our experiments with excess amounts of NP (mimicking the situation that occurs within infected cells). During our routine transfections, NP expression plasmids constitute 25% of the total DNA. NP does not become a limiting component in the assay until DNA levels are reduced to only 1-5% of the total (not shown). Thus, any minor variations in NP levels are unlikely to have an impact on polymerase activity in our system. As a demonstration of this, we used LiCOR Image Studio to quantify NP levels in the blot from the original submission, as well as two other replicates (Author response image 1). There was less than two-fold variation in NP levels across all experiments. Most conditions had expression levels very close to that of control NP levels (within ~20% of the control). In addition, we note that the minor changes in NP levels do not correlate with inhibition by PKCs. NP levels are marginally higher with expression of PKCβ2 and PKCη, yet overall polymerase activity is impaired in these settings (Figure 1). Similarly, in additional control experiments in Figure 1, NP levels are slightly higher in the presence of PKCs and polymerase activity is still impaired. While NP levels might look lower in the presence of PKCδ and PKCθ, total NP levels are actually above the control. This might be due in part to that fact that NP migrates as two distinct species with over 30% of NP present in the hyperphosphorylated form in these samples (Author response image 1 and Figure 1—figure supplement 1). As for the minor changes in Figure 2, if anything they reinforce our results – NP levels were slightly lower in cell lysates with PKCδ, PKCη and NP E339A, but these were the conditions that exhibited the strongest co-precipitation relative to their expression.

**Author response image 1. respfig1:** NP levels were quantified for data presented in Figure 1.

*Reviewer #2:*

*[…] The paper is well written and the conclusions are partially supported by the obtained data. Following open issues might be addressed:*

*1) The authors show that PKCδ phosphorylates NP to keep it in the monomeric form. As a result this protein is not available for the synthesis of full-length vRNA and cRNA. The transition from transcription to replication must be therefore regulated by an unknown phosphatase and not by PKCδ.*

We agree with the reviewer, and noted in the original manuscript that a phosphatase may be involved in licensing NP oligomerization. This possibility, along with others, is also included in the first and last paragraphs of the Discussion. Perhaps more to the point, we do not want to overstate the conclusions from our data and the role of PKCs in directly controlling the transition from primary transcription to genome replication. We have re-written sections throughout the manuscript to emphasize the role of PKC in establishing monomeric NP, and the role this balanced phosphorylation plays in regulating NP oligomerization and RNP assembly. We are careful to say that this enables the transition, but concurring with the reviewer’s point we take care not to claim that PKCδ drives the switch from transcription to replication.

*2) The authors speculate that PB2 bridges the binding of PKCδ to NP, however, a direct interaction between PKCδ and PB2 was not shown. Interestingly, PB2 itself seems to be phosphorylated by PKCδ based on the appearance of a higher migrating band (Figure 2). If true, PB2 might be a substrate of PKCδ and therefore co-purifies with this PKC isoform.*

NP and PKC stably interact only in the presence of PB2. We proposed that PB2 bridges interactions between NP and PKC, but as the reviewer suggests we cannot exclude the alternative that PB2 might co-precipitate via interaction with NP and not PKC. To test our model, we performed co-immunoprecipitation assays between PB2 and PKCδ in the absence of NP. PB2 specifically immunoprecipitated PKCδ, with only low levels of PKCδ present in the control pull down (Author response image 2). We have also begun to map this interaction using PB2 domain truncations, obtaining results that show PKCδ binding does not require regions in either the PB2 N- or C-terminus. These data show that NP is not required for PB2:PKC interactions, although they do not show that this interaction is direct as the reviewer implied (i.e. we cannot exclude that other cellular partners are present in the complex). The astute observation by the reviewer that PKCs might also target PB2 intrigued us as well. We are performing a completely separate line of experiments to characterize the interactions between PB2 and PKCδ and their functional consequences. This kind of fine-mapping of the interaction and extensive functional analyses of PB2-PKC interactions is interesting, but well beyond the scope of the current manuscript.

**Author response image 2. respfig2:** PB2 interacts with PKCδ in the absence of NP. PKCδ and PB2 variants were expressed in 239T cells.

*3) Phosphorylation of NP occurs independent on the presence of PB2 (Figure 2). Thus it is difficult to understand why PB2 should be important to bridge the interaction of PKCδ with NP.*

The reviewer correctly notes that PB2 is not required for NP phosphorylation. We also do not make this claim in the text, but instead note that PB2 stabilizes the association between PKC and NP, and that this complex is enriched for phospho-NP. We feel that this is more representative of the conditions during infection where both NP and PB2 will be present. We can envision multiple scenarios where this bridging might be important, possibly for controlling or maintaining NP phosphorylation, directing phosphorylation to only a subset of NP, facilitating phosphorylation of NP sites with suboptimal consensus recognition motifs, or it might be important in controlling the timing of NP phosphorylation. We cautiously speculate on these in the Discussion (last paragraph).

*4) The extend of NP phosphorylation in the presence of full length PKCδ seems to be negligible (e.g. Figure 2: no slower migrating form of NP in the Input lane). How does this translate to the inhibitory effect exerted by these kinases?*

We identified a hyper-phosphorylated form of NP that migrates slower in gels and can be increased by co-expressing certain PKCs. However, it is not known exactly which and how many modified sites are present in this hyperphosphorylation. It is not uncommon for anomalous migrations such as these to arise only after multiple phosphorylation events. This raised the possibility that our shifted protein represents NP modified at multiple positions, or may even require phosphorylation of specific residues. CIP treatment left behind an intermediate-shifted phospho-species, in agreement with the idea that hyperphosphorylated NP is a complex mix of phospho-species (Figure 1—figure supplement 1). We also performed new experiments using phosphate affinity SDS-PAGE. Phos-tag gels exaggerate mobility shifts due to phosphorylation. NP separated by conventional SDS-PAGE migrated as a single band and exhibited a single shifted species when PKCδ is co-expressed (Author response image 3). By comparison, NP separated on phos-tag gels revealed a new slower migrating species in the absence of PKCδ, and up to three distinct populations in the presence of PKCδ. These preliminary data show that the single band of NP resolved by conventional SDS-PAGE can contain phosphorylated forms of the proteins, and that the lack of a shifted NP species in conventional SDS-PAGE does not necessarily mean that NP phosphorylation is absent.

These data help explain why expression of full length PKCδ exhibits a functional effect on NP activity, but does not result in an obvious shift in mobility using conventional SDS-PAGE. In agreement with this conclusion, our quantitative mass spec shows that the knockout of PKCδ changes phosphorylation at multiple sites (Figure 5), even though NP recovered from these cells migrates as a single species. Thus, the presence of the hyperphosphorylated form is a useful proxy to demonstrate NP phosphorylation, but its absence cannot be used to conclude that NP is not phosphorylated or the relative abundance of any individual phosphorylation event. Further, as only a small fraction of NP in the infected cell is ever incorporated into an RNP, PKCδ might only need to phosphorylate this small sub-population of NP to exert an effect.

**Author response image 3. respfig3:** Phos-tag gels resolve multiple phospho-species of NP.

*5) Figure 3. The headline of the figure legend should be rephrased, since no experimental evidence was provided that PKC regulates RNP assembly.*

As noted above for reviewer 1 point 3, we have generated new data directly addressing this critique (Figure 3). Will have also changed the figure legend title and the header for this section of the Results (” PB2 stabilizes interactions between PKCδ and NP”).

*6) Figure 4: Without further purification of the vRNPs, it is unclear whether RNP-bound NP is analyzed or NP purified with PB2.*

We apologize for the confusion. The goal of this experiment was to test if active PKC co-purified with PB2 during infections and if the NP:PB2:PKC complexes we detected in transfected cells were functionally relevant in infected cells. It was not designed to assert the source of the NP that is modified in the kinase assay. The original submission referred to these as “PB2 complexes” to account for the presence of both binary PB2:NP, PB2:PKC:NP, PB2:PKC and even RNPs. We continued this inclusive phrasing in the resubmission. We have included additional data demonstrating that PKCδ is activated during influenza virus infection (Figure 4—figure supplement 1). Phospho-proteomic analysis identified peptides from PKCδ containing phosphorylated S302 and S304 (Figure 4—figure supplement 1 and in the subsection “Activated PKCδ associates with the viral polymerase during infection”). These phosphorylations are known to be autocatalytic and serve as a marker of PKCδ activation (Durgan et al., 2007), further supporting the claim of Figure 4 that activated PKCδ co-purifies with the viral RNP during infection.

*7) Figure 6: There is a substantial effect on entry in both the PRKCD^-/-^-1 and -2 cells (panel A, which might significantly affect viral growth efficiency. Moreover, replication seems to be affected too at early time points post infection. It is however unclear which block might contribute most to the observed growth deficits.*

We agree with the reviewer that our data suggest PKCδ plays roles both during entry and after entry, and that these composite effects will be represented in replication efficiency. We now make this explicit in the last paragraph of the subsection “NP phosphorylation and influenza virus replication are impaired in PKCδ-deficient cells”. It is not clear which of these activities might be more important, but we were excited by the possibility that PKCδ might be important for multiple steps. This possibility was a major motivation for performing the step-wise analysis in Figure 6, which confirmed roles at both entry and post-entry steps. We have since acquired more data to demonstrate that PKCδ affects viral gene expression post-entry. We previously showed that influenza gene expression is impaired in *PRKCD^-/-^*cells infected with VSV-G pseudotyped virus that bypasses HA-dependent entry (Figure 5). As a control, we performed additional experiments to test if VSV-G-mediated entry requires PKCδ (Figure 5—figure supplement 2 and subsection “NP phosphorylation and influenza virus replication are impaired in PKCδ-deficient cells”, fourth paragraph). WT and knockout cells inoculated with VSV-G pseudotyped influenza virus or *bona fide* VSV and analyzed by flow cytometry. The number of infected cells was indistinguishable between WT and knockout cells, for both pseudotyped influenza virus and VSV. These data show that PKCδ is not required for VSV-G-mediated entry and that VSV-G successfully bypasses any PKCδ-dependent entry effect for influenza virus. They provide more confidence for our claim that PKCδ plays an important role post-entry where it regulates the RNA replication machinery. As we then show in Figure 6, PKCδ does indeed impact HA-mediated entry.

*8) Figure 6: Interference was also measured in the so-called "Replication assay" which was initiated by an acid bypass. In this very artificial assay it is unclear how the vRNPs are actually transported to the cell nucleus and whether PKCδ affect this transport step.*

The acid bypass assay we used was developed in the early 1980s by Kai Simons and Ari Helenius and was key in showing that influenza virus enters via endocytosis (Maitlin, et al. 1981 JCB) This assay cleanly separates entry from post-entry events. It was more recently used to show that viral uncoating involves the aggresome (Banerjee, et al. 2014 Science). More germane here, experiments by Banerjee, et al. also revealed that viral particles that enter by acid bypass resume use of the “normal” entry pathway requiring similar cellular partners as canonical influenza entry. These references are now included in our Materials and methods (subsection “Virus replication and entry assays”, second paragraph). Our data also suggest that PKCδ is not required for nuclear import of incoming RNPs. We detect similar levels of vRNA and mRNA in the presence of cycloheximide for both WT and knockout cells, suggesting equivalent stability and transport of the vRNP into the nucleus (Figure 6).

*Results provided in panel C might suggest that primary transcription in the presence of CHX is not affected. Quantification of triplicates are required to draw this conclusion. In addition, primary transcription of viral mRNA should be determined after "normal" infection of wt and PRKCD^-/-^-1 or -2 cells in the presence of CHX.*

These experiments are now quantified in triplicate and presented in Figure 6. They still show that PKCδ does not affect primary transcription. The reviewer suggests measuring viral mRNA following “normal” entry to test for decreases in viral transcription. We have already addressed this with data showing that PKCδ is important for HA-dependent entry (Figure 6), and that defects in gene expression still exist when we bypass this process with pseudotyped virus (Figure 5). This approach also avoided concerns about the acid bypass technique (Figure 5).

[Editors' note: the author responses to the re-review follow.]

*[…] Scientific comments:*

*1) While the presented data are very solid and the interpretations leave no obvious alternate explanations, the apparently opposite roles of PKC in early transcription and subsequent replication of the viral genomes, via phosphorylation of NP monomers and modulation of NP oligomerization need to be better discussed. While the authors point out some of these conundrums, a figure should be provided that models the phosphorylation/oligomerization of NP in transcription and replication.*

To help convey the conclusions of the paper and the ultimate implications, we have included a model figure (Figure 7). Both the figure and the associated legend synthesize the data and many of the arguments made in the paper into our current working model. It also highlights implications of this model and areas still open for more investigation. We note that the manuscript already discussed that PKCδ is important during genome replication, but that it is dispensable for primary transcription (Figure 6, subsection “PKCδ regulates viral genome replication”, last paragraph). This is not necessarily an opposing function as suggested by the reviewer. Rather, our data indicate that PKCδ simply does not play a dominant role in primary transcription because there is no need to regulate NP oligomerization at this step – primary transcription occurs on pre-formed vRNPs where NP oligomerization has already occurred. The experimental rationale and results have been clarified in the first and last paragraphs of the subsection “PKCδ regulates viral genome replication”.

*2) Putative phosphatases that are postulated to be involved if PKC phosphorylates NP late in infection could be discussed. It would be interesting to comment on any evidence that PKC substrate specificity is dynamic during infection.*

This is a keen observation. To further emphasize these possibilities, we included that “Protein phosphatase 6 was reported to directly bind multiple subunits of the viral polymerase and regulate RNA synthesis during infection, although it is not known if it targets any of these proteins for dephosphorylation (York et al., 2014). It is also possible that PKCd activity or target specificity changes throughout infection to impact NP oligomerization potential.” As we do not have definitive data on whether regulation is performed by actively removing phosphates from NP or passively reducing NP phosphorylation by changing PKC activity, we are hesitant to more strongly promote these ideas or one versus the other.

*3) In general, a probe for the component actually subjected to the immunoprecipitation in the post IP fraction would be nice, otherwise it is hard to determine if the observed lack of enrichment (or surplus of enrichment) is due to differential immunoprecipitation or bone-fide differences in affinity. This becomes particularly a concern owing to different PKC isoforms serving as the target for IP rather than a single target with interaction partners varying. Presumably, those data have been obtained or can be obtained. Some reassurance that recovery of the targeted protein is consistent, and comparable between the iso forms of PKC, should be provided.*

We have included new data addressing this concern. While we were precipitating different PKC isoforms, they all contained the same epitope tags and were expected to behave similarly in an IP. We now show that all of the PKC isoforms are immunoprecipitated equivalently relative to their expression in the cell lysate. Representative data from three replicates are shown in the IP sections of Figure 2. This is also reflected in the first paragraph of the subsection “PB2 stabilizes interactions between PKCδ and NP” and a change to Figure 2 legend.

*4) Figure 3. Is it known what other kinase target sites are observed in panel D (S165A/S470A)? Can this be discussed? Why does PKC2 not allow the oligomerization of GFP-TL and NPΔTL? Clearly, it can phosphorylate NPs.*

NP is phosphorylated on at least 10 positions, including 3 sites in the tail loop (S402, S403 and S407) (Hutchinson, et al. 2012, Mondal, et al. 2015). *In vitro* assays in Figure 3 directly addressed only S165 and S407. We suspect the remaining phosphorylation that occurs on the S165/407 double mutant is likely lower frequency modifications at these other sites. Supporting this conclusion, modification is reduced at some of these other site, especially S402 and S403, in cells lacking PKCδ (Figure 5). It should be noted that mutation of S402 or S403 did not impair polymerase function (Hutchinson 2012, Mondal 2015), and were thus not explored further here. We now note this in the third paragraph of the subsection “PKC phosphorylates NP at the tail loop:groove interface and blocks oligomerization”.

We thank the reviewers for pointing out the interesting results with PKCβ2. This is something we have considered as well, but do not have a definite answer. Substrate specificity can be relaxed in *in vitro* kinase assays, which is why we felt it important to confirm functional impacts of these observation in cell-based assays that better represent the native conditions during infection. This caveat was included in the original text and has been expanded briefly in the last paragraph of the subsection “PKC phosphorylates NP at the tail loop:groove interface and blocks oligomerization”.

*5) Figure 4. Has the oligomerization status of NP and associated PB2 before and after the kinase assay? This would reveal whether RNP-associated NP can be phosphorylated. As a monomer or oligomer?*

This would be an additional way to test out model, but we have not performed the suggested experiment and it would be very technically challenging. However, our data in Figure 4 show that RNP-associated NP can be phosphorylated – NP in immunopurified RNPs is a substrate for PKC, although with low efficiency. In addition, phospho-proteomics on virions, where NP is packaged as part of RNPs, also identified phosphorylated sites on NP (Hutchinson, et al. 2012). Thus, our data suggest NP in an RNP can be phosphorylated, albeit inefficiently.

*Comments about presentation:*

*1) In Figure 2, panels A, C, and D have bands that are clearly overexposed. While we understand the difficulty in displaying data with a wide dynamic range, including additional, less-exposed, panels would allow for better comparisons between samples.*

We agree that it is a challenge to capture the linear range of protein levels by western blot, especially for protein present at very different levels. Our blots were performed to detect even the lowest interactors or expressors. We revisited the blots from these experiments, and while we have multiple exposures from 4 separate experiments, unfortunately we do not have any very short exposures. This is why we are very careful to only make qualitative or relative claims about binding, and not quantitative measures.

*Panel D also has a far under-loaded immunoprecipitation control relative to the experimental samples.*

In some experiments, NP expression can be affected by PKC co-expression, resulting in the lower levels in the control on the left side of Figure 2 relative to conditions with PKC. This condition showed no background precipitation. In addition, in the case where NP levels were higher, we still detected very little co-precipitation with PKCβ2, confirming the specificity of our IP under both low and high NP levels. This was further verified with the specific IPs performed with full-length PKC on the right side of the panel.

*2) The figures are dense and even though the text follows the data closely, additional clarity such as the addition of lane numbers would greatly improve the exposition.*

There is a significant amount of data presented, and we worked carefully to design and label the figures. We feel that adding lane numbers would be counter to the reviewer’s desires, making the figures busier and the text more confusing. These can be added if absolutely necessary.

*3) Additional clarity would be provided if Figure 1 (i.e. Polymerase assays) and 3 (i.e. RNP assembly assays) had top diagrams as in Figure 6.*

These suggestions have been included as supplemental figures associated with each figure.

*4) In the last paragraph of the subsection “PB2 stabilizes interactions between PKCδ and NP”, the NP:PB2:PKCδ complex is termed a heterotrimer. While this may be stoichiometrically correct with respect to these components, higher-order versions or other interaction partners could be involved. Demonstrating this is unnecessary for the present manuscript and a simple change in the language to hetero-oligomer would be more than sufficient.*

We agree and have corrected the text.